# Pseudotachylyte as field evidence for lower crustal earthquakes during the intracontinental Petermann Orogeny (Musgrave Block, Central Australia)

Friedrich Hawemann[1], Neil S. Mancktelow[1], Sebastian Wex[1], Alfredo Camacho[2], Giorgio Pennacchioni[3]

1 Department of Earth Sciences, ETH Zurich, Sonneggstrasse 5, CH-8092 Zurich
2 Department of Geological Sciences, University of Manitoba, 125 Dysart Rd, Winnipeg, Manitoba, R3T 2N2 Canada.
3 Department of Geosciences, University of Padova, Via Gradenigo 6, 35131 Padova, Italy

*Correspondence to*: Friedrich Hawemann (friedrich.hawemann@erdw.ethz.ch)

**Abstract.** Geophysical evidence for lower continental crustal earthquakes in almost all collisional orogens is in conflict with the widely accepted notion that rocks, under high grade conditions, should flow rather than fracture. Pseudotachylytes are remnants of frictional melts generated during seismic slip and can therefore be used as an indicator of former seismogenic fault zones. The Fregon Subdomain in Central Australia was deformed under dry sub-eclogitic conditions of 600-700 °C and 1.0-1.2 GPa during the intracontinental Petermann Orogeny (ca. 550 Ma) and contains abundant pseudotachylyte. These pseudotachylytes are commonly foliated, recrystallized, and crosscut by other pseudotachylytes, reflecting repeated generation during ongoing ductile deformation. This interplay is interpreted as evidence for repeated seismic brittle failure and post- to inter-seismic creep under dry lower crustal conditions. Thermodynamic modelling of the pseudotachylyte bulk composition gives the same P-T conditions of shearing as in surrounding mylonites. We conclude that pseudotachylytes in the Fregon Subdomain are a direct analogue of current seismicity in dry lower continental crust.

## 1 Introduction

Predicting the rheology of the Earth's crust is crucial for all geodynamic models over the whole range of length and time scales from plate tectonics to seismic hazard estimation. In general, the main constraints on rock rheology are derived from rock deformation experiments, with results obtained at high strain rates and high temperatures extrapolated to natural conditions (e.g. Kohlstedt et al., 1995). The simplest assumption of competing brittle and viscous behaviour at constant strain rate results in a typical "Christmas-tree" 1D representation of strength variation with depth (Goetze and Evans, 1979). One basic form of the strength profile for the continental lithosphere is the so-called "jelly sandwich" model, with a quartz- and feldspar-rich, wet, weak, and viscously flowing lower crust sandwiched between a strong brittle upper crust and a dry, strong, brittle upper mantle with olivine rheology (e.g. Burov and Watts, 2006; Jackson, 2002a). An alternative "crème brûlée" model considers a wet olivine rheology for the upper mantle, and therefore limits all significant strength and seismicity to the upper crust (Burov and Watts, 2006; Jackson, 2002a). However, in contradiction to such models that limit brittle deformation exclusively to the upper crust, seismicity is also recorded in the lower crust in almost all collisional settings, e.g. the Alps (Deichmann and Rybach, 1989; Singer et al., 2014), the Himalayas (Jackson, 2002b; Jackson et al., 2004), the Tien Shan (Xu et al., 2005), the central Indian shield (Rao et al., 2002), and the North Island of New Zealand (Reyners et al., 2007).

The main factors governing rock rheology are temperature, strain rate, chemical composition, water activity, and pore fluid pressure. These parameters cannot be well constrained from seismic measurements. Consequently, direct observations from field studies of exposed lower crustal sections are critical for understanding lower crustal rheology. Pseudotachylytes, generally interpreted to represent frictional melt generated during seismic failure (McKenzie and Brune, 1972; Sibson, 1975), can be locally abundant in exposures of lower crust (Altenberger et al., 2011, 2013; Austrheim and Boundy, 1994; Clarke and Norman, 1993; Moecher and Steltenpohl, 2009, 2011; Pittarello et al., 2012; Orlandini et al., 2013; Menegon et al., 2017). The metamorphic conditions of these sections correspond to depths well below the usual brittle-ductile transition zone for crustal rocks (<15 km) and thus the assumed lower limit for earthquake nucleation. Sibson (1980) reported mutually overprinting pseudotachylytes and mylonites from the Outer

Hebrides Thrust (NW Scotland) and similar observations were made by Moecher and Steltenpohl (2009) and Menegon
et al. (2017) in the Lofoten region (N Norway), by Hobbs et al. (1986) in the Redbank Shear Zone (Arunta Block,
Central Australia), and by Camacho et al. (1995) in the Woodroffe Thrust (Central Australia). Mutual overprinting
has been interpreted to reflect the generation of pseudotachylytes and mylonitization under the same conditions
(Altenberger et al., 2011, 2013; Clarke and Norman, 1993; Moecher and Steltenpohl, 2011; Pennacchioni and Cesare,
1997; Pittarello et al., 2012; Ueda et al., 2008; White, 1996, 2004, 2012). A possible explanation for the generation of
earthquakes in these mid- to lower crustal rocks is the downward migration of the brittle-ductile transition through the
transfer of stress from the upper crust after a seismic event (Ellis and Stöckhert, 2004; Handy and Brun, 2004). Another
mechanism for the embrittlement of the lower crust are high pore fluid pressures, and many field examples of
pseudotachylytes and brittle fracturing in the lower crust have been closely linked to fluid activity (Altenberger et al.,
2011; Austrheim et al., 1996; Lund and Austrheim, 2003; Maddock et al., 1987; Steltenpohl et al., 2006; White, 2012).
In contrast, Clarke and Norman (1993) considered that the preservation of fine-grained pseudotachylyte under high
grade conditions is only possible if the pseudotachylyte composition is dry.
The aim of the current study is to establish the conditions under which pseudotachylytes can form in a water deficient
lower crust and to demonstrate that the recurring interplay of fracture and flow represents the bulk deformation style
of lower crust in intracontinental settings as preserved in the Musgrave Block. The field, petrological and
microstructural results provide direct observational constraints on proposed models for lower crustal seismicity.
**2 Geological Setting**
The Musgrave Block in Central Australia (Fig. 1) provides excellent exposure of well-preserved lower crustal fault
rocks that can be studied over hundreds of kilometres (Figs. 2a,b). In this study, we focus on the Fregon Subdomain
in the eastern Musgrave Block, which represents the hanging wall of the Woodroffe Thrust (Camacho et al., 1997;
Camacho and McDougall, 2000; Wex et al., 2017).
The Fregon Subdomain experienced granulite facies
metamorphism during the Musgravian Orogeny,
associated with the amalgamation of the Australian
Cratons at about 1.2 Ga (Gray, 1978; Wade et al.,
2006). The voluminous Pitjantjatjara Supersuite,
consisting mainly of granites and charnockites, was
emplaced during the post-collisional stage (Smithies
et al., 2011). Extension at ~1070 Ma is manifested
by the intrusion of dolerite dykes (Alcurra Suite),
gabbros, and granites (Giles event; Evins et al.,
2010). This rift event does not seem to be associated
with a deformation phase in the eastern Musgrave
Block, and was probably purely magmatic (Aitken

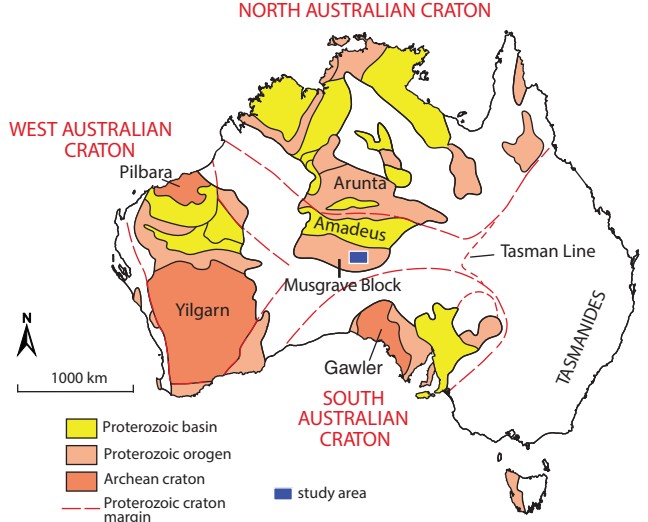

**Figure 1: Position of the Musgrave Block between the Archean cratons of Australia. Modified after Evins et al. (2010).**

et al., 2013). Another series of dolerite dykes, the Amata Suite at ca. 800 Ma, is potentially related to a mantle plume
(Zhao et al., 1994). The Fregon Subdomain preserves a series of mostly strike-slip, crustal-scale shear zones developed
during the Petermann Orogeny (~550 Ma; Camacho et al., 1997), all of which are associated with abundant
pseudotachylytes. During the Petermann Orogeny, the Fregon Subdomain was juxtaposed against former mid-crustal
rocks in the north (Mulga Park Subdomain) along the moderately to shallowly south-dipping Woodroffe Thrust (Fig.
2; Camacho et al., 1995; Major and Conor, 1993; Wex et al., 2017). The intracontinental Petermann Orogeny correlates
in time with the global Pan-African Orogeny (Camacho et al., 1997) and was possibly caused by a clockwise rotation
of the South and West Australian Cratons with respect to the North Australian Craton (Li and Evans, 2011). The
protoliths of the Fregon and Mulga Park Subdomains are very similar in composition and age (Camacho and Fanning,
1995; Edgoose et al., 1999), but can be readily distinguished using airborne thorium (Th) concentrations as seen in
Fig. 2c. The low Th concentration in the hanging wall probably relates to the formation and migration of partial melts
to shallower crustal levels during the earlier granulite facies metamorphism, with the breakdown of apatite and
monazite resulting in partitioning of incompatible elements, such as Th, into the melt phase (Förster and Harlov,
1999). Consequently, low Th concentrations can be used to indicate that the crust experienced granulite facies
metamorphism (Lambert and Heier, 1968; Scharbert et al., 1976). The signal is partly obliterated by the granitic
intrusions of the Pitjantjatjara Supersuite and Giles Event, which succeeded granulite facies metamorphism.

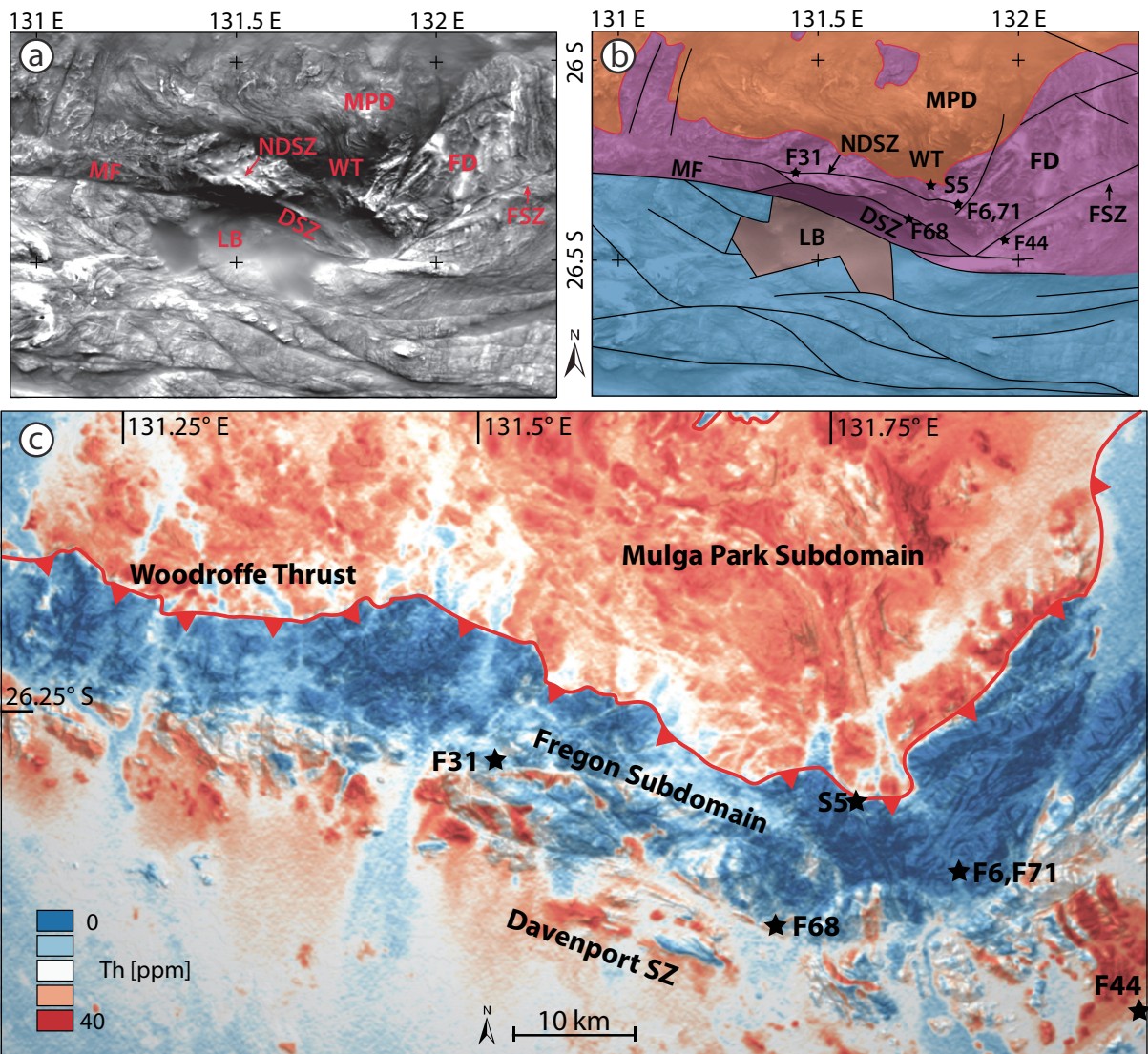

**Figure 2: a) Total magnetic intensity map (Milligan and Nakamura, 2015) and interpreted structures. Most fault zones appear as dark lines with a marked contrast, lithological layering is visible in the Mulga Park Subdomain (MPD), whereas the sediments of the Levenger Basin (LB) appear blurred. b) Interpretation of the tectonic framework of the Central Musgrave Block. The Mann Fault (MF) separates units that did not experience high grade overprint during the Petermann Orogeny in the south (blue), from the Fregon Subdomain (FD, purple) in the north. The Davenport Shear Zone (DSZ), North Davenport Shear Zone (NDSZ) and the Woodroffe Thrust (WT) were mapped by integrating the magnetic intensity map with airborne imagery and direct field observations. c) Compilation of airborne gamma ray surveys, with concentration of thorium shown from blue (low) to red (high). Flares of low concentration in the footwall are associated with sediments transported from the hanging wall by rivers. Pseudotachylyte sample locations discussed in the text are indicated as black stars. Dataset SA_RAD_TH, Geological Survey of South Australia (2011), grey levels from hill shade. For a simplified geological map covering the same area, and an interpreted synthetic NS cross-section, see Wex et al. (2017).**


## 3 Field observations

The Davenport Shear Zone (DSZ) is a strike-slip shear zone trending WNW-ESE with a sub-horizontal stretching lineation, a moderately to steeply dipping foliation (Camacho et al., 1997), and a sense of shear that changes from dominantly sinistral to dextral from west to east, reflecting the regional variation in the foliation trend. In the framework of the Musgrave Block, the DSZ is bounded to the south by the generally poorly exposed Mann Fault (Fig. 2a). While dextral strike-slip movement along the Mann Fault is indicated by the pull-apart Levenger Basin (Aitken and Betts, 2009; Camacho and McDougall, 2000), a normal, north-side up component is implied by the lack of any known high-pressure Petermann Orogeny overprint south of the Mann Fault, as inferred from the mapping work of Glikson et al. (1996), the age data of Camacho and McDougall (2000), and our own observations. To the north, deformation in the DSZ is strongly partitioned and bounded by a high-strain zone. The only continuous zone of mylonites north of the DSZ towards the Woodroffe Thrust is the coeval North Davenport Shear Zone (NDSZ) (Camacho et al., 1997). This mylonitic zone developed under similar conditions to the DSZ, but the pitch of the lineation is widely variable, from horizontal to down dip to the south, with the shear sense being dominantly dextral-oblique thrusting towards NW. The DSZ mylonites and the NDSZ converge to the west. Towards the east, the relationships are less clear because of the lack of outcrop. The ENE trending, moderately dipping Ferdinand Shear Zone is a sinistral strike-slip shear zone that appears to branch from the steep Mann Fault.

The DSZ is an approximately 5 km wide mylonitic zone with the foliation trend clearly visible on satellite images. This foliation encompasses low strain domains, from kilometre to metre scale, which potentially preserve initial stages of the temporal development of deformation. Pseudotachylytes are abundant, not only in the DSZ, but throughout the whole Fregon Subdomain. They are concentrated along, but not exclusively limited to, the different shear zones described above and especially along the Woodroffe Thrust (Camacho et al., 1995). Pseudotachylytes are easily identified in the field by their aphanitic matrix, abundance of clasts, injection veins, breccias and chilled margins (Fig. 3). When overprinted by subsequent ductile shearing, identification becomes more difficult and cannot always be confirmed (Kirkpatrick and Rowe, 2013; Price et al., 2012). The thickness of pseudotachylyte veins reaches up to 7 cm but is usually about 1 cm. Generation surfaces, when observed, show very little former melt, as it was mostly injected into the host rock. There is no evidence for hydration, such as formation of bleached halos or hydrous mineral growth. Assemblages with significant amounts of water-bearing minerals (e.g. biotite and hornblende) are restricted to late- to post-Musgravian granitic intrusions. The pseudotachylytes do not show any specific affinity for these more hydrous units, but in fact occur in all lithologies. In all the different mylonitic shear zones of the Fregon Subdomain, the observed relative age relationship between pseudotachylyte formation and ductile shearing in the adjacent rock covers the following range of possibilities.

(1) Pseudotachylyte post-dates shearing. The mylonitic foliation in the host rock is in general crosscut and brecciated by the pseudotachylyte (Fig. 3a), although the pseudotachylyte may also be emplaced parallel to the foliation, in some cases at the boundary to ultramylonite bands or along the rim of dolerite dykes (Fig. 3b). Pseudotachylytes occur as veins or as breccias with a black aphanitic matrix, in which fragments of the host rock show a rotated internal fabric.

(2)  Pseudotachylyte is broadly synchronous with shearing. Pseudotachylyte veins crosscut the mylonitic foliation

136           and are themselves foliated, as visible from elongated clasts (Fig. 3c). The stretching lineation in the

137           pseudotachylyte is parallel to that in the surrounding mylonites. Veins and breccias can show a wide range

138           of matrix colours, from grey to beige to caramel-coloured.

(3)  The pseudotachylyte itself is foliated but occurs in effectively unsheared rocks, with ductile deformation

140           confined to the pseudotachylyte vein, while the adjacent rock remained little deformed (Figs. 3d, 4).

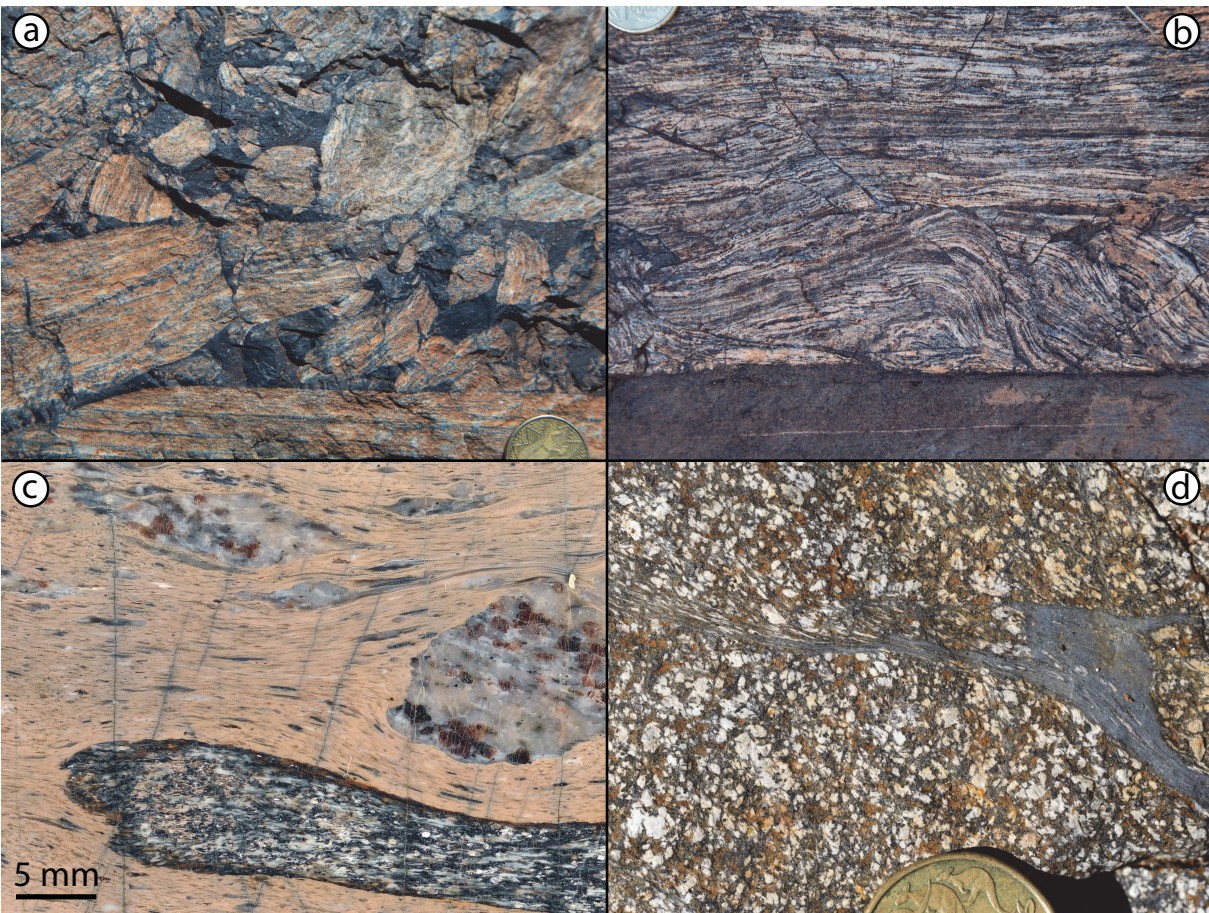

**Figure 3: Field examples of pseudotachylytes: (a) Pseudotachylyte breccia disrupting mylonitic foliation. Note the relative rotation of clasts, their generally angular shape and the wide range of clast sizes. (26.3877 S, 131.7091 E). (b) Late-stage pseudotachylyte localizing at the boundary of a sheared dolerite dyke (bottom part of the image), creating a duplex-like structure with all planes of movement decorated by pseudotachylyte (N is up, 26.3408 S, 131.5255 E). (c) Polished slab with caramel-coloured pseudotachylyte including fragments of quartzo-feldspathic gneiss and mafic granulite. Note the internal foliation and elongation of clasts. Note also that although the clasts are variably foliated, they are not ultramylonitic and have irregular shapes and a very wide range of sizes, typical of a cataclastic breccia (26.3853 S, 131.7105 E). (d) Sheared pseudotachylyte in an otherwise almost undeformed gabbro (N is up, 26.3528 S, 131.8419 E).**


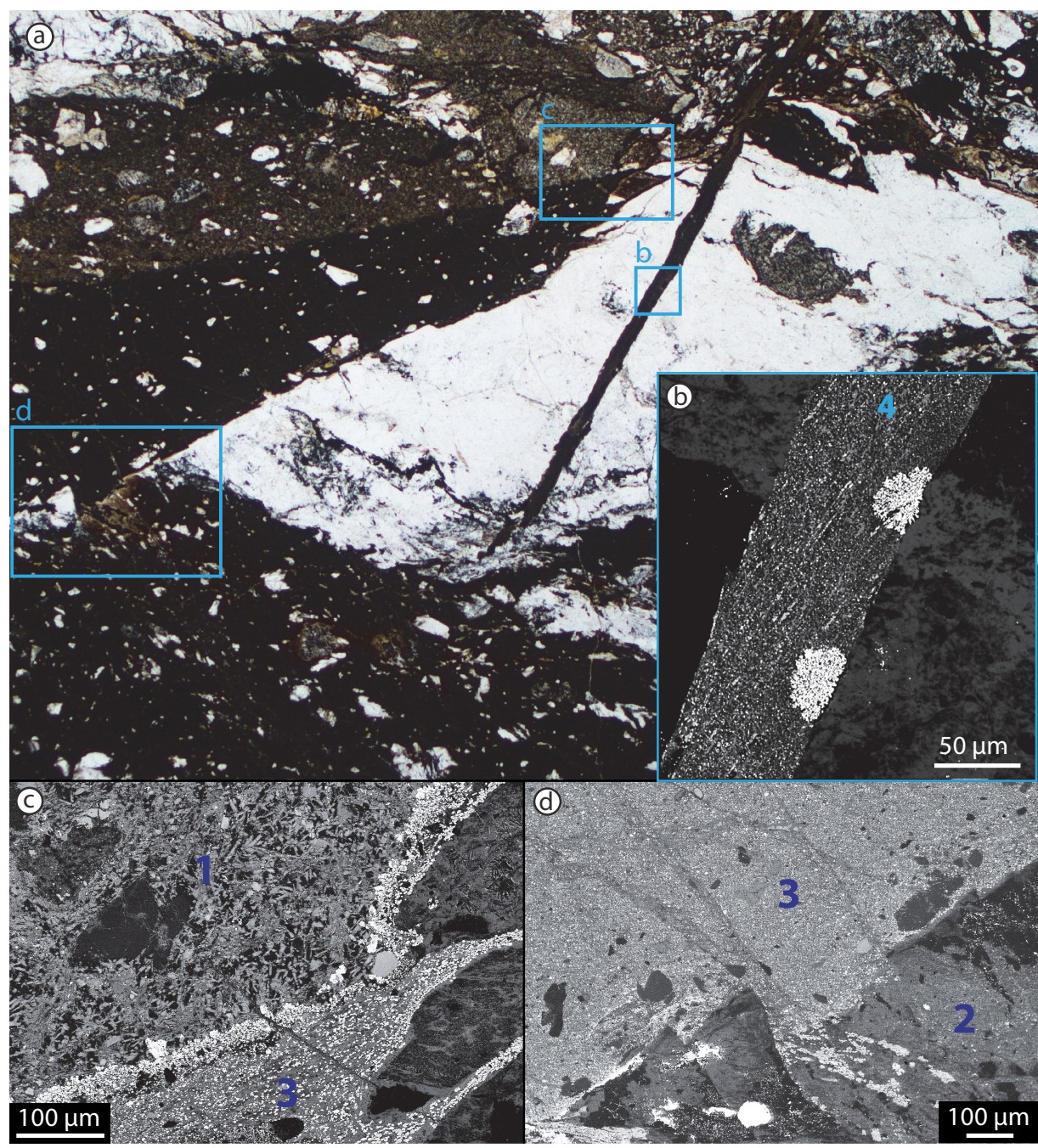

**Figure 4: a) Thin section image of sample F44 (26.4514 S, 131.9553 E) showing four generations of pseudotachylyte, from oldest 1 to youngest 4 (plane polarized light; to reduce contrast, images taken with different exposure times were combined). b) Backscattered electron image of the area indicated by the blue box in (a): a vein of generation 4 shows a planar foliation, defined by elongate clasts, that is oblique to the vein boundary and is overgrown by dendritic (or "cauliflower") garnet. c) Pseudotachylyte generations 1 and 3 showing a former chilled margin now decorated by garnet. d) Pseudotachylyte generation 3 crosscutting generation 2, with the boundary offset by late fractures.**


## 4 Microstructure

### 4.1 Post-shearing pseudotachylyte

Late-stage pseudotachylytes crosscut the mylonitic fabric, and show the pristine characteristic microstructures of quenched melts, preserving an extremely fine-grained matrix (grain size on the order of a few microns) with flow structures, chilled margins and angular, sometimes corroded clasts of iron oxides (Fig. 5a). In some instances, microlites of feldspar and pyroxene are observed. Since these microlites are the result of crystallization during melt undercooling, their mineral assemblages and mineral chemistry do not represent ambient temperature conditions. Al-rich pyroxenes have been described from pseudotachylytes in the Musgrave Ranges some 250 km west of the current study area by Wenk and Weiss (1982). Pressures obtained from the geobarometers applied were about 3 GPa, which the authors interpreted to represent dynamic syn-pseudotachylyte melting pressures, rather than ambient lithostatic conditions.

### 4.2 Syn-shearing pseudotachylyte

Sheared pseudotachylytes on occasion contain clasts of an older generation of pseudotachylyte, suggesting recurring brittle and ductile deformation. The syn-kinematic mineral assemblage of pseudotachylytes does not show any evidence for fluid infiltration.

Sample F31, located in the North Davenport Shear Zone (26.2793S, 131.4968 E), is from the immediate boundary between a garnet-bearing quartzo-feldspathic gneiss and a dolerite dyke. This contact is exploited by a pseudotachylyte (Fig. 6), which mostly incorporates the dolerite but also includes clasts of the felsic gneiss. In addition to inclusions of country rock, there are also clasts of an older generation of pseudotachylyte, strongly overprinted by ductile shear, within the breccia. Locally, the boundary of these first generation pseudotachylyte clasts is marked by a second, also sheared, generation of pseudotachylyte of greyish colour that crosscuts the older generation but is itself cut by the younger unsheared third generation (Fig. 6). These relationships demonstrate that (1) initial pseudotachylyte formation, interpreted to represent a brittle seismic event, was followed by (2) ductile shearing, followed by (3) a second seismic event, developing the grey second generation pseudotachylyte, which was then (4) again sheared, to be finally followed by (5) a third generation of unsheared pseudotachylyte and associated breccia.

Sample F68 is a garnet-bearing quartzo-feldspathic gneiss, sampled close to the northern boundary of the DSZ (same outcrop as the example in Fig. 3c; 26.3849 S, 131.7067 E). Pseudotachylyte veins are ca. 1 mm thick, spaced ca. 1 cm apart, and oriented parallel to the proto-mylonitic foliation. Pseudotachylyte veins show injections and have a fine-grained matrix of Grt+Kfs+Pl+Qz+Bt+Ky+Rt, similar to the host rock assemblage, where Ky is restricted to Pl-clasts (mineral abbreviations are after Whitney and Evans, 2010). The pseudotachylyte is slightly enriched in Bt relative to the host rock, but no other OH-bearing phases are present. Kyanite was identified by using Raman spectroscopy and EBSD. The fine grained poikilitic garnet (~20 μm, Fig. 5b) results in the caramel colour in the field (Fig. 3c). The internal foliation is defined by biotite and aggregates of garnet (Fig. 5b). In the host rock, mm-sized relict, granulite facies garnets are fractured and surrounded by smaller, neocrystallized garnet, with sizes on the order of tens of microns.

A sheared pseudotachylyte was sampled in the immediate hanging wall of the Woodroffe Thrust (sample S5, 26.3082
S, 131.7745 E), at the boundary between a sheared dolerite dyke and undeformed felsic granulite. This
pseudotachylyte has a paragenesis similar to the dolerite dyke (Pl+Cpx+Gt+Ky+Rt+Ilm+Qz+Kfs), but is much finer
grained. The boundary with the dolerite is decorated by even finer grained garnet, possibly the remnant of a chilled
margin with a slightly different composition. Where the pseudotachylyte injected into the granulite, it evaded shearing
and shows a fine-grained matrix with dendritic garnet overgrowth (Fig. 5c), possibly directly crystallizing form the
melt. The original flow banding is highlighted by the preferential overgrowth of garnet on some bands, probably due
to compositional differences (Fig. 5d).
**4.3 Sheared pseudotachylyte in undeformed host rock**
Sample F44 from the Ferdinand Shear Zone (26.3856 S, 131.9550 E) contains at least four generations of
pseudotachylyte veins and breccias developed in a granitic host rock (Fig. 4a-d). Individual pseudotachylyte veins
vary in the amount and rounding of clasts, compositional heterogeneity, and the mineral assemblage. The modal
abundance of Grt+Cpx+Opx+Amp+Fsp is also variable, possibly reflecting a progressive change in bulk chemistry of
the melt. The observed mineral assemblages in this unsheared pseudotachylyte might either be the result of
crystallisation directly from the melt, or later static overgrowth. Generation four clearly crosscuts older generations
and shows an internal foliation, which is interpreted to be the result of a ductile overprint rather than flow banding, as
it is nearly planar with a consistent oblique angle to the margin of the pseudotachylyte. The margin of this
pseudotachylyte is decorated with dendritic garnet that clearly overgrows the planar foliation (Fig. 4b), indicating
post-shearing high grade conditions rather than crystallization from the melt.
Sample F6 is a gabbro assigned to the Giles Complex (Fig. 3d; 26.3528 S, 131.8419 E), which largely preserves its
magmatic texture, but contains sheared pseudotachylyte. The host rock is almost undeformed and shows static
reactions such as Grt coronas around Pl in contact with Cpx and breakdown of Opx and Pl to Cpx. The pseudotachylyte
contains a large number of clasts (ca. 50% of the total volume), dominantly of Pl, which show limited recrystallization.
The matrix minerals of the dynamically recrystallized pseudotachylyte consists of
Grt+Cpx+Kfs+Qz+Mag+Rt+Ilm+Ky.

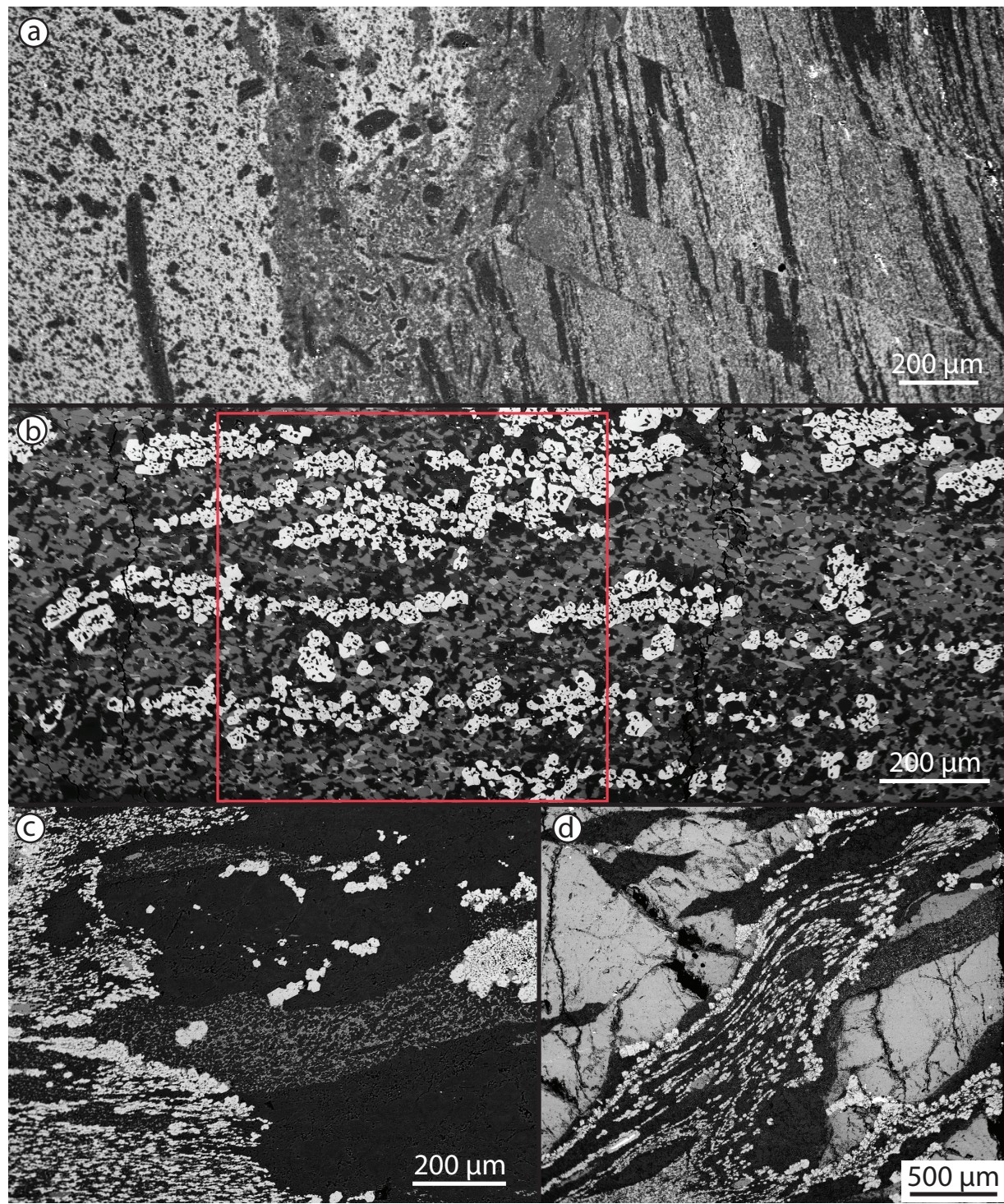


**Figure 5: Backscattered electron images of pseudotachylyte: (a) Late-stage pseudotachylyte with angular clasts in mylonitic host rock with abundant fractures (26.3550 S, 131.8432 E). (b) Dynamically recrystallized pseudotachylyte in sample F68. Minerals in greyscale from dark to bright are Qz, Pl, Kfs, Ky, Bt, Grt. Red box indicates the mapped area for Fig. 7. (c) Unsheared pseudotachylyte in a vein cutting through a plagioclase grain of the granulitic host rock showing dendritic overgrowth of garnet. In the left part of the image, the pseudotachylyte is fine grained and foliated (sample S5). d) Injection vein preserves original flow banding, visible through the selective overgrowth of garnet (sample S5).**

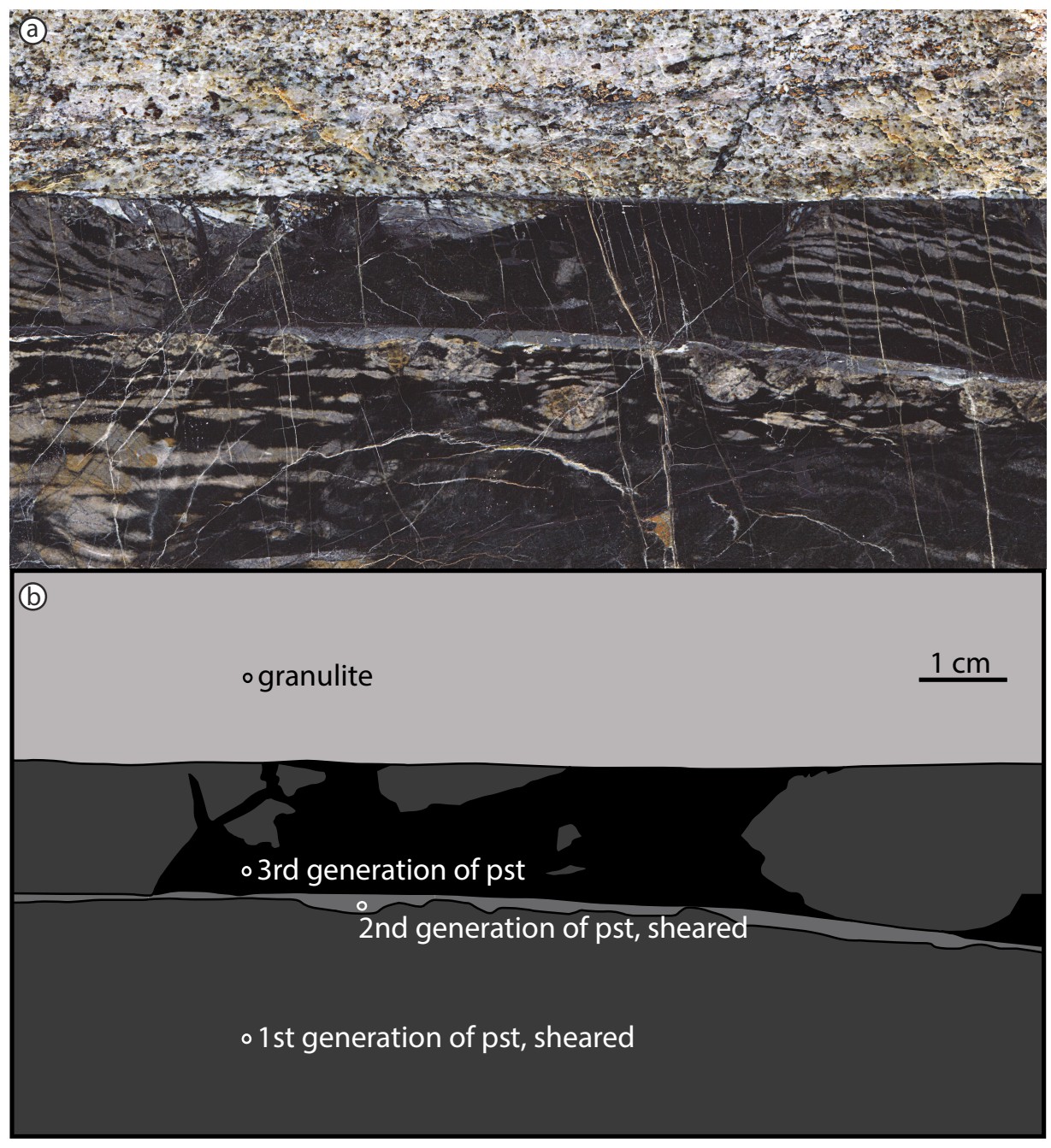


**Figure 6: a) Scan of a polished rock slab (sample F31; 26.2793S, 131.4968 E), and b) sketch of the same area. The sample shows three generations of pseudotachylyte, developed at the boundary between garnet-bearing quartzo-feldspathic gneiss (to the top) and a dolerite dyke (below and outside the image). The first generation of pseudotachylyte contains clasts of the quartzo-feldspathic host (upper part of the image), which are intensively sheared. This generation is crosscut by a second generation of pseudotachylyte, which is present as a light grey vein, with much smaller clasts, which are also elongated. The third generation of pseudotachylyte exhibits a sharp boundary to the host rock in the upper part of the image and incorporates clasts of the first generation pseudotachylyte.**

## 5 Conditions of pseudotachylyte emplacement

### 5.1 Methods

Backscattered electron (BSE) images were taken with a FEI Quanta 200F scanning electron microscope, equipped with a field emission gun deployed at the ScopeM (Scientific Center for Optical and Electron Microscopy, ETH Zurich). Quantitative measurements of mineral composition were acquired with a JEOL JXA-8200 electron probe micro analyser (EPMA) at the Institute for Geochemistry and Petrology, ETH Zurich, with a set of natural standards. Voltage was reduced from 15 kV to 10 kV for some samples to account for the fine grain size. Thermodynamic modelling using Perple_X (Connolly, 1990) was carried out on three samples of recrystallized pseudotachylytes within different host rocks. The determination of a bulk composition for pseudotachylytes by using the classic XRF-method (X-Ray Fluorescence) is hampered by their geometry and the presence of abundant clasts (Di Toro and Pennacchioni, 2004). To minimize these problems, the Matlab toolbox XMapTools (Lanari et al., 2014) was used to calculate the bulk composition from WDS-maps (wavelength dispersive spectrometer) collected with the EPMA. Quantitative point analysis was used to "standardize" the maps (Lanari et al., 2014). Here, the weight per cent (wt%) of a point analysis is linked to counts for each element of the same point on the map. This can be done for each mineral phase separately to account for matrix effects. After correlating the counts to wt% of all pixels, the bulk composition of the pseudotachylyte for the desired area of the map was extracted and used as input for Perple_X. For all samples, a standardization for each separate mineral was impossible because of the fine grain size. Instead, all count values on the map were correlated to a mean wt% value from point analysis. The resulting deviation in mineral chemistry is generally low and was corrected manually by comparing exported compositions from the standardized maps with measured analyses. The bias on the bulk composition induced by the choice of area can be tested by using a Monte Carlo approach (integrated in XMapTools). The deviations in wt% are in the order of 0.4 for silica and much lower for the other elements. The thermodynamic dataset of Holland and Powell (1998) was used to calculate pseudosections for the composition of the samples and a range of P-T-conditions to compare with the observed assemblage in dynamically recrystallized pseudotachylyte. The solution models used can be found in the appendix (Tables B1-3).

### 5.2 Results

#### 5.2.1 Syn-shearing pseudotachylyte

The pseudotachylyte veins in sample F68 have a homogeneous phase distribution with a relatively large grain size (~20 µm), and are almost devoid of clasts (Fig. 5b). The compositional (WDS) map, which was used for calculation of a pseudosection (Fig. 7), has a size of 400x400 pixels and measurements were made using a step size of 2 µm, resulting in an area of 0.64 mm$^2$. The amount of water in the rock could not be measured directly, and was calculated

using an assumption of 3 wt% water in biotite and its modal
abundance, since biotite is the only OH-bearing mineral. As
biotite is a platy mineral, its area in the section parallel to the
lineation and perpendicular to the foliation might be under-
represented. However, an arbitrary threefold increase of bulk
water content in the calculations (from 0.05 to 0.15 wt%)
does not have a noticeable effect on the stability fields of the
mineral phases. The stability field for the assemblage of the
recrystallized pseudotachylyte in sample F68 is wide, which
is why pressure-temperature (P-T) conditions were further
delimited with mineral isopleths (Fig. 7). The conditions
estimated are around 1.05 GPa and 600 °C. The
stoichiometry for each mineral can be reliably reproduced
(Table B1).

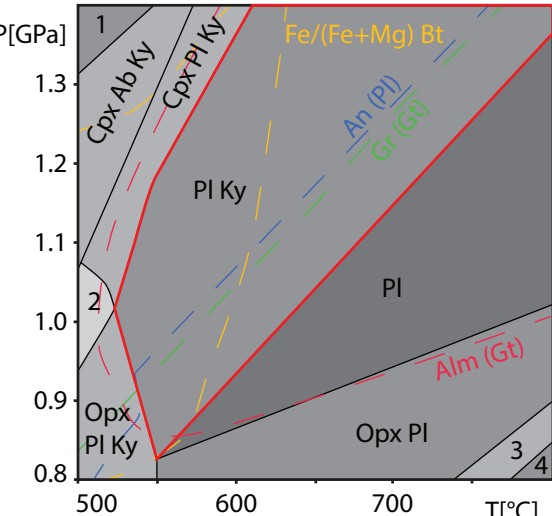

**Figure 7: Pseudosection calculated for F68. Additional phases in all fields: Kfs+Grt+Bt+Qz+Rt. With isopleths for Fe/(Fe+Mg) in biotite, anorthite component of plagioclase (An (Pl)), grossular and almandine component of garnet (Gr, Alm (Gt)). Numbered Fields: 1: Cpx, Ky; 2: Opx, Cpx, Pl, Ky; 3: Opx, Pl, Ilm; 4: Opx, Pl, Ilm, no Rt**

In sample S5, the pseudotachylyte shows strong
compositional heterogeneity parallel to the foliation,
probably due to differences associated with original flow
banding. This is best visible in the Ca-compositional map of Fig. 8a, where areas 1 and 2 show lower Ca-content in
Pl with respect to the other areas. Areas 1, 2 and 3 have a similar paragenesis of Grt+Cpx+Pl+Kfs+Rt, with Qz limited
to area 2, while area 3 also lacks Kfs. Areas 4 and 5 consist of Grt+Cpx+Pl+Bt+Opx+Rt. A bulk composition was
calculated individually for each area. Clasts of Ca-rich Pl are present (see upper right corner of 7a for an example),
with Ky needles growing inside the clasts but not in the matrix assemblage. These Pl-clasts were masked out for the
calculation of the local bulk composition since they are not part of the stable assemblage. Calculated pseudosections
for each area were superimposed onto each other to narrow down the P-T estimates of coeval formation (Fig. 8b).
Area 4 was not considered, since modelling predicted sapphirine to be stable, which was not observed in the sample.
Otherwise, the stable assemblage field for area 4 overlaps largely with those of the other areas. The stability of Opx
with the bulk compositions of areas 4 and 5 is limited to a maximum pressure of about 0.8 GPa. Since Opx occurs as
coronas around Cpx, we assume that Opx-growth is post-kinematic (see area 5 in Fig. 8a, where Opx appears as small
dark blue dots around the Cpx). Therefore, Opx was not considered to be stable in the sheared paragenesis of area 5.
The pseudosections show an overlap of the different stable parageneses for their respective local bulk composition
(Fig. 8b). The shared stability field spans the range 1.1-1.3 GPa and 670-710 °C. The compositions of individual
phases derived from the Perple_X model, calculated at 1.2 GPa and 690 °C, are in good agreement with the measured
compositions (Table B2).

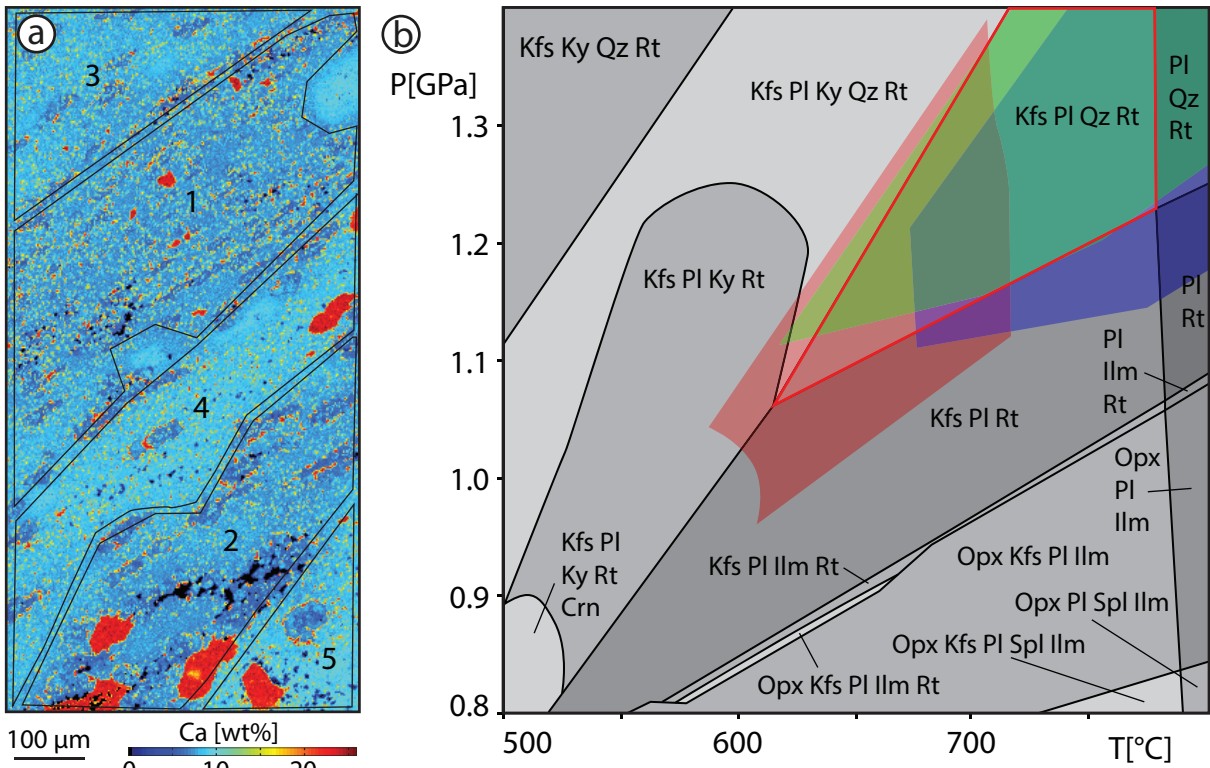

**Figure 8: Quantified X-ray map for Ca for sample S5 with a step size of 2 μm and 250x500 pixels. Minerals visible: red: Cpx, dark blue: Grt, medium blue: low-Ca Pl, light blue: high-Ca Pl. Areas are defined by the Pl-composition. b) Pseudosection for sample S5, area 2; all parageneses also have Grt+Cpx. Overlays of the observed stability fields for parageneses from pseudosections from the other areas: red: area 1, green: area 3, blue: area 5. For the microstructural context of the area, see Fig. A1.**

### 5.2.2 Sheared pseudotachylyte in undeformed host rock

Pseudotachylyte in the gabbroic sample (F6) is extremely fine grained and is dominated by millimetre-sized clasts of Pl, which only partly reacted to form Grt and Kfs. The compositional (EDS) map was collected with a step size of 1 μm and 400x500 pixels, to account for the small grain size. The area is located between a remnant Pl-clast, overgrown by Grt with the rim replaced by Kfs, and a ribbon of mixed Kfs and Pl (Figs. 9a,c). The area in between, with abundant Grt+Ap+Mag, is interpreted to have directly originated from the former pseudotachylyte melt and recrystallized during shearing. Smaller Fsp-clasts were masked out during determination of the local bulk composition because reactions and mixing seem to be incomplete. Apatite was removed completely for the calculation of the composition, as P was not measured nor integrated into the modelling. The high content of $Fe^{3+}$-bearing minerals, such as Ilm and Mag (Fig. 9c), required that the $Fe^{2+}/Fe^{3+}$ ratio to be calculated using the volume per cent of each iron-bearing phase and their respective $Fe^{2+}/Fe^{3+}$ ratio. The calculated pseudosection (Fig. 9b) shows a narrow area for the observed assemblage of Grt+Cpx+Kfs+Qz+Ilm+Rt+Mag+Ky at conditions of ca. 1.23 GPa and 590 °C. Rt only appears as exsolution lamellae from the Ti-rich Ilm, which is a reaction taking place close to the P-T conditions derived from pseudosection modelling. Initial calculations were done with the Cpx solution model used for the other samples, resulting in lower

pressures (ca. 1.15 GPa), but predicted much higher Na-content in the Cpx of 6.5 wt% compared to the measured 2
wt%. The Cpx-model used for the final calculations yields compositions much closer to those measured (Table B3).

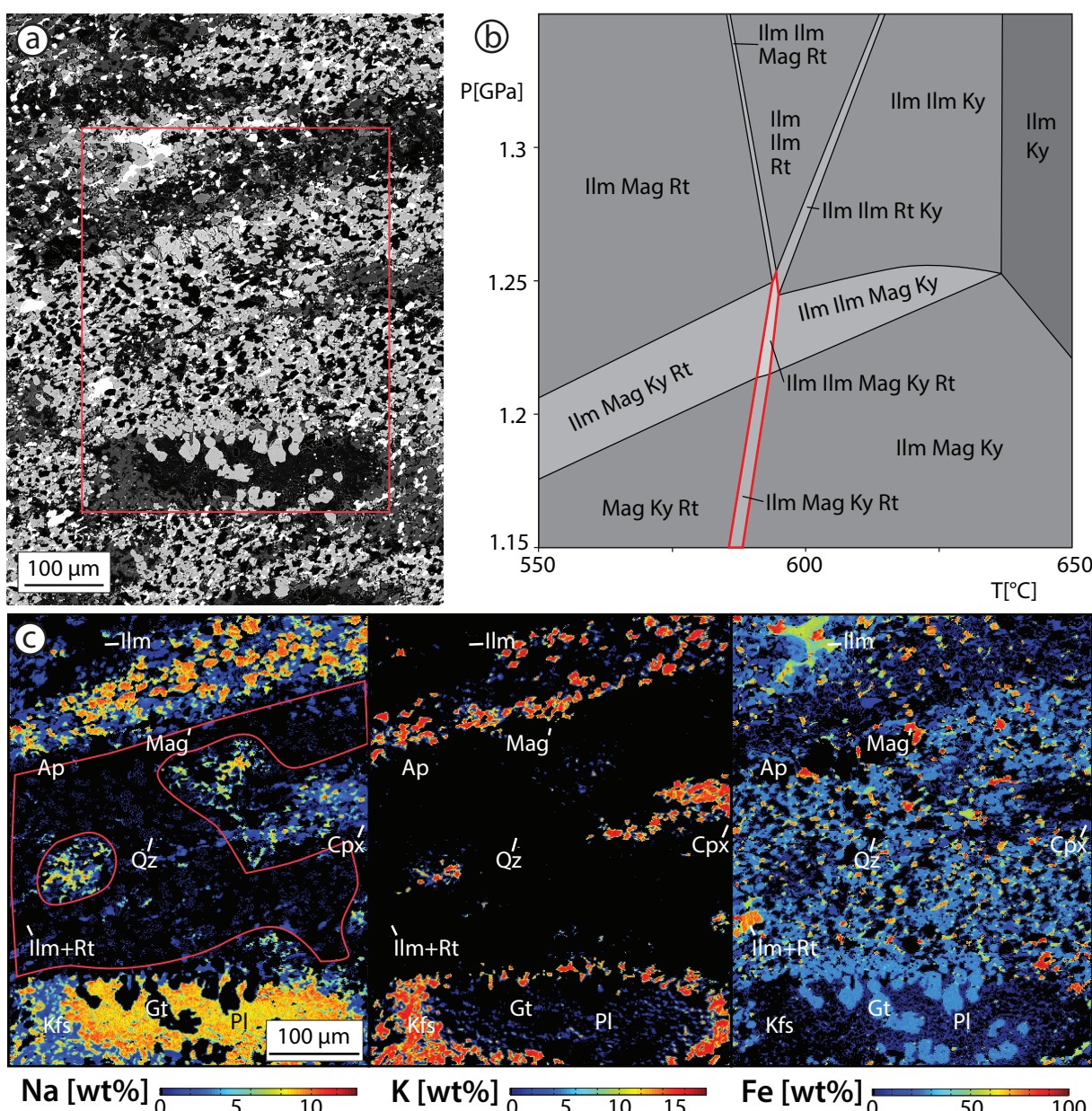


**Figure 9: a) BSE image of a sheared pseudotachylyte with partly recrystallized clasts. Red box indicates the location of the X-ray-map. b) Results from thermodynamic modelling using Perple_X with an estimate for the conditions of shearing at about 1.23 GPa and 590 °C. Minerals stable in all fields: Grt, Cpx, Kfs, Qz. c) Compilation of X-ray maps: Na-map shows the incomplete breakdown of a Pl-clasts in the bottom of the image and the replacement with Kfs (K-map). Red outline shows the extracted area of the bulk composition used. Fe-map shows abundant Mag (red) and two distinct Ilm populations (green and yellow).**

**6 Summary**

Multiple crosscutting sheared pseudotachylytes can be interpreted as a repeated interplay between brittle and ductile deformation. As a general observation, alternating seismic fracture and aseismic creep could potentially involve even more cycles, but it becomes increasingly difficult to demonstrate, because periods of accumulated shear strain localized on the pseudotachylyte zones tend to obscure earlier crosscutting relationships. Based on this clear evidence for repeated interplay, the pressure and temperature conditions derived from the dynamically recrystallized assemblage of sheared pseudotachylyte are interpreted to be close to the ambient host rock conditions of pseudotachylyte formation and injection. Thermodynamic modelling results yield values of 1.0-1.3 GPa and 600-700 °C. These results are very similar to the estimated conditions of mylonitisation in the Fregon Subdomain during the Petermann Orogeny of 650 °C and 1.2 GPa (Ellis & Maboko, 1992; Camacho et al., 1997). Such metamorphic conditions during the Petermann Orogeny imply an average geothermal gradient of ca. 16-18 °C/km for the studied rocks, as already noted by Camacho et al. (1997) and Wex et al. (2017) in the current area and by Scrimgeour and Close (1999) in the Mann Ranges further to the west. These values are low in comparison to those typical of collisional orogens and are more characteristic of cratonic continental crust (Sclater et al., 1980). Indeed, as discussed by Wex et al. (2017), measured heat flow values in region of the Musgrave Block would imply similar values for the geothermal gradient in the middle to lower crust today.

Lin et al., (2005) described pseudotachylytes in the hanging wall of the Woodroffe Thrust and interpreted them to have been generated during Musgravian Orogeny granulite facies metamorphism. This interpretation can be ruled out for two main reasons: 1) The hanging wall of the Woodroffe Thrust experienced granulite facies metamorphism during the ca. 1.2 Ga Musgravian Orogeny but all pseudotachylytes observed in the field and described in Lin (2005) are associated with structures related to the ca. 550 Ma Petermann Orogeny. 2) Pseudotachylytes are present in gabbros (Fig. 3d) and dolerite dykes (Fig. 3b) that intruded during the ca. 1.07 Ga Giles Event and dolerite dykes of the ca. 800 Ma Amata Suite. All these magmatic rocks were intruded well after the granulite facies metamorphism associated with the Musgravian Orogeny.

**7 Discussion**

Pseudotachylyte development by brittle failure and frictional seismic slip (McKenzie and Brune, 1972; Sibson, 1975) is the favoured mechanism to explain the field observations in the Fregon Subdomain. Alternative processes involving thermal runaway during ductile shear (John et al., 2009; Thielmann et al., 2015) or ductile instabilities (Hobbs et al., 1986) require that a pseudotachylyte-bearing fault necessarily had a ductile precursor. This is not in accord with the observation that many pseudotachylytes occur in otherwise unsheared host rocks and act as a precursor for subsequent ductile shearing, rather than the other way around. In addition, pseudotachylytes within undeformed host rock do not necessarily contain clasts of mylonites and especially not clasts of ultramylonites. The clasts in pseudotachylytes are also typically angular and show a very wide size range (Figs. 3-6), which is consistent with fracture and brecciation. As discussed above, there can be repeated cycles of pseudotachylyte formation and shearing, with the result that clasts of sheared pseudotachylyte are included in later pseudotachylyte. This very fine grained, sheared material is preserved

and not totally consumed by melting. It cannot therefore, be argued that all evidence for a precursor ultramylonitic
zone is lost because the ultramylonite is always totally melted during subsequent "self-localizing thermal runaway"
(John et al, 2009). We would argue that examples such as shown in Fig. 6, where the pseudotachylyte zone discretely
crosscuts an older granulite facies foliation at a low angle without any evidence for crystal-plastic shearing, is best
explained by seismic fracture and pseudotachylyte development by frictional melting. Furthermore, fractured garnet
is potentially an indicator for seismic stresses (Trepmann and Stöckhert, 2002) and has been reported to occur
specifically in close association with pseudotachylytes (Austrheim et al., 2017). However, in the area of the current
study, fracturing of older granulite facies garnet is widespread and not limited to the immediate border of
pseudotachylytes.
Brittle deformation under elevated temperatures at depths below the classic brittle-ductile transition zone in felsic
continental crust might be explained by local high fluid pressure promoting fracturing (Altenberger et al., 2011; Lund
and Austrheim, 2003; Steltenpohl et al., 2006; White, 2012), either due to dehydration reactions or fluid infiltration.
However, these mechanisms can be excluded for the examples presented here, because most host rocks (in particular
the felsic granulites) were already thoroughly dehydrated during the earlier granulite facies Musgravian Orogeny and
there is no evidence for fluid infiltration during the Petermann Orogeny. As seen for example in sample S5, the hydrous
mineral biotite is restricted to isolated domains, indicating that the activity of OH was low. The absence of hydration
associated with pseudotachylyte development in the shear zones described here also indicates that the switch between
seismic brittle fracture (pseudotachylyte) and ductile shearing was not induced by infiltration of fluids. This is in
marked contrast to what has been previously described in the Bergen Arc (Austrheim, 2013, and references therein)
and Lofoten area (Menegon et al., 2017) of Norway, where fluid influx promoted by propagation of the earthquake
fracture and associated weakening due to metamorphic reaction localized subsequent ductile shearing. In the absence
of elevated pore fluid pressure, high stresses are necessary to fracture rocks under dry, lower crustal conditions (Sibson
and Toy, 2006; Menegon et al, 2017). Natural examples of shear zones with small grain sizes developed under
amphibolite facies conditions suggest that mid- and lower crust can be strong (Fitz Gerald et al., 2006; Menegon et
al., 2011). This might explain initial fracturing, but on the long term, shear zones show localization of strain and
therefore indicate weakening of the rocks. To explain the observed cyclicity of fracture and flow, temporal stress
variations are necessary. Transient high stresses in the mid- to lower crust have been proposed to result from a
downward propagation of stresses from the usual seismogenic zone (<15 km) during seismic failure (Ellis and
Stöckhert, 2004a; Handy and Brun, 2004; Moecher and Steltenpohl, 2009). In the example of the 2015 Gorkha
earthquake on the Main Himalayan Thrust (Duputel et al., 2016), there are indeed aftershocks located in the deeper
crust following an earthquake at about 15 km depth. Alternatively, for a "jelly-sandwich" style lithospheric model,
stress propagating upwards from the seismogenic zone in the strong upper mantle could also explain lower crustal
seismicity. Both of these options are hard to test from field observations. However, the implication of these conceptual
models is that for each event recorded in the lower crust (> 30 km depth), such as the pseudotachylytes in the
Davenport Shear Zone, there was necessarily a large earthquake with a source in the upper crust or upper mantle.
However, this is not observed for many large, lower crustal earthquakes, for example in the Indian Shield (Mitra et
al., 2004). Considering the abundance of pseudotachylytes in the lower crustal Fregon Subdomain, this would imply
a correspondingly large and perhaps unrealistic amount of strong seismicity in the upper crust or upper mantle
respectively, suggesting that such localized pseudotachylytes may have had a local trigger in the dry lower continental
crust.

**8 Conclusions**

The Fregon Subdomain documents seismic fracturing under lower crustal conditions of about 1.0-1.3 GPa and 600-
700 °C in an intracontinental setting. Repeated episodes of brittle failure and ductile creep represent recurring
earthquake cycles and a strong variation of stress in a water-deficient lower crust. It is questionable whether current
models of downward propagation of seismic stresses from the "seismogenic" upper crust can explain the observed
repetition of brittle failure and ductile shearing sporadically distributed over such a wide area. It seems more likely
that these earthquake cycles are locally triggered in the dry lower continental crust, at least in this intracontinental
setting. Models should therefore take into account temporal and spatial variations of stress in a heterogeneously
deforming lower crust.

**Appendix A, additional images**

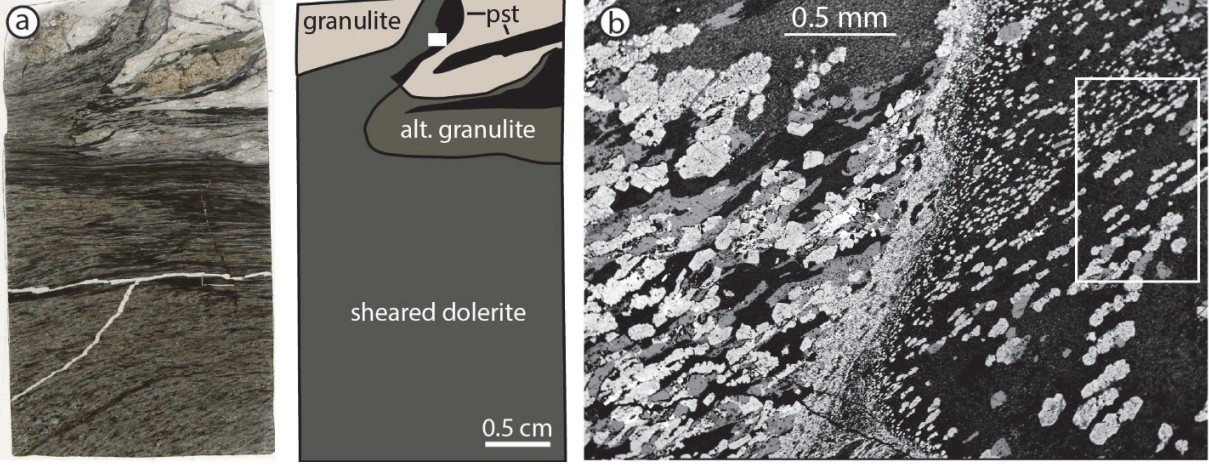

**Figure A1: Microstructural context of area mapped in sample S5 (Fig. 8): a) Plane polarized light microscopic image and**
**sketch of the thin section, with box indicating image in b). b) BSE image of the boundary between dolerite (left) and sheared**
**pseudotachylyte, with the white box indicating area in Fig. 8a.**

**Appendix B, Bulk and mineral chemistry**

| | Bulk | Grt_m | Grt_c | Pl_m | Pl_c | Kfs | Kfs_c | Ky_m | Ky_c | Bt_m | Bt_c |
|---|---|---|---|---|---|---|---|---|---|---|---|
| $Na_2O$ | 1.06 | 0.02 | 0.00 | 8.77 | 8.14 | 0.89 | 1.33 | 0.00 | 0.00 | 0.19 | 0.00 |
| MgO | 2.67 | 8.31 | 9.28 | 0.01 | 0.00 | 0.00 | 0.00 | 0.00 | 0.00 | 19.04 | 18.56 |
| $Al_2O_3$ | 12.76 | 22.61 | 22.30 | 21.93 | 24.20 | 18.92 | 18.56 | 62.40 | 62.92 | 14.73 | 17.67 |
| $SiO_2$ | 70.2 | 38.55 | 39.42 | 58.69 | 61.49 | 63.10 | 65.08 | 36.66 | 37.08 | 37.61 | 37.61 |
| $K_2O$ | 3.77 | 0.02 | 0.00 | 0.19 | 0.55 | 15.53 | 14.91 | 0.00 | 0.00 | 10.19 | 10.76 |
| CaO | 1.98 | 5.83 | 5.63 | 5.40 | 5.61 | 0.05 | 0.12 | 0.03 | 0.00 | 0.01 | 0.00 |
| $TiO_2$ | 0.79 | 0.08 | 0.00 | 0.04 | 0.00 | 0.01 | 0.00 | 0.04 | 0.00 | 3.97 | 4.75 |
| MnO | 0.23 | 0.92 | 0.88 | 0.01 | 0.00 | 0.00 | 0.00 | 0.05 | 0.00 | 0.00 | 0.01 |
| FeO | 5.85 | 23.72 | 22.48 | 0.11 | 0.00 | 0.19 | 0.00 | 1.12 | 0.00 | 7.36 | 7.60 |
| $H_2O$ | 0.05* | | | | | | | | | 3** | 3.04 |
| total | 99.31 | 100.06 | 100.00 | 95.15 | 99.99 | 98.69 | 100.00 | 100.29 | 100.00 | 96.10 | 100.00 |
| | | | | | | | | | | | |
| **Cations** | | | | | | | | | | | |
| Al | | 2.04 | 2.00 | 1.21 | 1.26 | 1.04 | 1.01 | 1.99 | 2.00 | 1.29 | 1.48 |
| Si | | 2.96 | 3.00 | 2.75 | 2.73 | 2.94 | 2.99 | 0.99 | 1.00 | 2.79 | 2.68 |
| | | 5.00 | 5.00 | 3.96 | 3.99 | 3.98 | 4.00 | 2.98 | 3.00 | 4.08 | 4.16 |
| Fe | | 1.52 | 1.43 | | | | | | | 0.46 | 0.45 |
| Mg | | 0.95 | 1.05 | | | | | | | 2.11 | 1.97 |
| Mn | | 0.06 | 0.06 | | | | | | | | |
| Ca | | 0.48 | 0.46 | 0.27 | 0.27 | 0.00 | 0.01 | | | | |
| Na | | | | 0.80 | 0.70 | 0.08 | 0.12 | | | | |
| K | | | | 0.01 | 0.03 | 0.92 | 0.87 | | | 0.96 | 0.98 |
| | | | | | | | | | | | |
| total | | 3.01 | 3.00 | 1.07 | 1.00 | 1.00 | 1.00 | | | 3.53 | 3.40 |

**Table B1: Representative analysis for sample F68. m=measured; c=calculated from Perple_X at 1.2 GPa and 690 °C; *calculated: volume per cent Bt and 3 weight per cent water in Bt; **assumed; Solution models: Omph(GHP), GlTrTsPg, melt(HP), Chl(HP), Sp(HP), Gt(GCT), Opx(HP), Mica(CHA1), Ctd(HP), St(HP), Bio(TCC), hCrd, Osm(HP), Carp(HP), Sud, feldspar, IlGkPy, Neph(FB), Chum**

|  | Area 1 | Area 2 | Area 3 | Area 4 | Area 5 | Grt_m | Grt_c | Pl_m | Pl_c | Kfs_m | Kfs_c | Cpx_m | Cpx_c |
|---|---|---|---|---|---|---|---|---|---|---|---|---|---|
| Na$_2$O | 4.42 | 3.89 | 5.59 | 5.60 | 4.72 | 0.00 | 0.00 | 6.71 | 6.97 | 0.19 | 1.47 | 1.23 | 1.47 |
| MgO | 4.79 | 5.37 | 2.56 | 2.44 | 4.16 | 11.52 | 11.25 | 0.06 | 0.00 | 0.09 | 0.00 | 15.71 | 15.05 |
| Al$_2$O$_3$ | 21.00 | 21.15 | 23.50 | 23.87 | 22.33 | 23.41 | 22.71 | 25.97 | 25.62 | 19.01 | 18.72 | 3.76 | 2.42 |
| SiO$_2$ | 52.30 | 52.04 | 54.99 | 55.45 | 53.85 | 40.16 | 40.14 | 59.33 | 59.30 | 64.04 | 64.94 | 51.85 | 55.20 |
| K$_2$O | 0.58 | 0.71 | 0.42 | 0.47 | 0.49 | 0.01 | 0.00 | 0.28 | 0.82 | 14.29 | 14.58 | 0.01 | 0.00 |
| CaO | 8.76 | 9.02 | 8.76 | 9.58 | 10.12 | 7.26 | 7.26 | 7.44 | 7.30 | 0.17 | 0.29 | 22.25 | 23.10 |
| TiO$_2$ | 0.41 | 0.36 | 0.38 | 0.35 | 0.33 | 0.03 | 0.00 | 0.10 | 0.00 | 0.04 | 0.00 | 0.20 | 0.00 |
| MnO | 0.15 | 0.16 | 0.12 | 0.11 | 0.12 | 0.35 | 0.52 | 0.01 | 0.00 | 0.00 | 0.00 | 0.02 | 0.00 |
| FeO | 5.12 | 5.92 | 2.30 | 1.13 | 2.47 | 18.17 | 18.12 | 0.13 | 0.00 | 0.45 | 0.00 | 3.75 | 2.78 |
| H$_2$O | 0.00 | 0.00 | 0.00 | 0.00 | 0.00 |  |  |  |  |  |  |  |  |
| total | 97.53 | 98.62 | 98.62 | 99.00 | 98.58 | 100.91 | 100.00 | 100.02 | 100.00 | 98.28 | 100.00 | 98.78 | 100.00 |


| Cations |  |  |  |  |  | Grt_m | Grt_c | Pl_m | Pl_c | Kfs_m | Kfs_c | Cpx_m | Cpx_c |
|---|---|---|---|---|---|---|---|---|---|---|---|---|---|
| Al |  |  |  |  |  | 2.04 | 2.00 | 1.36 | 1.35 | 1.04 | 1.01 | 0.16 | 0.10 |
| Si |  |  |  |  |  | 2.97 | 3.00 | 2.64 | 2.65 | 2.98 | 2.99 | 1.92 | 2.00 |
|  |  |  |  |  |  | 5.01 | 5.00 | 4.01 | 4.00 | 4.03 | 4.00 |  |  |
| Fe |  |  |  |  |  | 1.12 | 1.13 |  |  |  |  | 0.12 | 0.08 |
| Mg |  |  |  |  |  | 1.27 | 1.25 |  |  |  |  | 0.87 | 0.81 |
| Mn |  |  |  |  |  | 0.02 | 0.03 |  |  |  |  | 0.00 | 0 |
| Ca |  |  |  |  |  | 0.58 | 0.58 | 0.36 | 0.35 | 0.01 | 0.01 | 0.88 | 0.90 |
| Na |  |  |  |  |  |  |  | 0.58 | 0.60 | 0.02 | 0.13 | 0.09 | 0.10 |
| K |  |  |  |  |  |  |  | 0.02 | 0.05 | 0.85 | 0.85 | 0.00 | 0.00 |
| total |  |  |  |  |  | 2.99 | 3.00 | 0.95 | 1.00 | 0.87 | 1.00 | 1.95 | 1.90 |

**Table B2: Representative analysis for sample S5. m=measured; c=calculated from Perple_X at 1.2 GPa and 690 °C, all**
**mineral chemistry from area 2; Solution models: Omph(GHP), GlTrTsPg, melt(HP), Chl(HP), Sp(HP), Gt(GCT), Opx(HP),**
**Mica(CHA1), Ctd(HP), St(HP), Bio(TCC), hCrd, Osm(HP), Carp(HP), Sud, feldspar, IlGkPy, Neph(FB), Chum**

| | Bulk | Gt_m | Gt_c | Kfs_m | Kfs_c | Cpx_m | Cpx_c |
|---|---|---|---|---|---|---|---|
| $Na_2O$ | 0.17 | 0.01 | 0.00 | 0.67 | 0.10 | 1.99 | 2.35 |
| MgO | 3.76 | 6.27 | 5.90 | 0.04 | 0.00 | 11.96 | 10.18 |
| $Al_2O_3$ | 11.75 | 21.88 | 21.31 | 19.54 | 18.36 | 4.02 | 6.76 |
| $SiO_2$ | 53.22 | 38.79 | 38.54 | 62.81 | 64.75 | 52.46 | 49.90 |
| $K_2O$ | 0.04 | 0.00 | 0.00 | 15.97 | 16.75 | 0.03 | 0.00 |
| CaO | 5.09 | 6.64 | 7.35 | 0.08 | 0.04 | 20.15 | 20.88 |
| $TiO_2$ | 3.24 | 0.10 | 0.00 | 0.04 | 0.00 | 0.25 | 0.00 |
| MnO | 0.38 | 0.92 | 0.82 | 0.01 | 0.00 | 0.09 | 0.00 |
| FeO | 14.90 | 26.43 | 25.37 | 0.69 | 0.00 | 9.07 | 3.84 |
| $Fe_2O_3$ | 7.87 | | 0.77 | | | | 6.08 |
| $H_2O$ | 0.00 | | | | | | |
| total | 100.41 | 101.04 | 100.05 | 99.84 | 100.00 | 100.01 | 99.99 |
| | | | | | | | |
| Cations | | | | | | | |
| Al | | 1.99 | 1.96 | 1.07 | 1.00 | 0.18 | 0.30 |
| Si | | 2.99 | 3.00 | 2.93 | 3.00 | 1.95 | 1.87 |
| | | 4.98 | 4.96 | 4.00 | 4.00 | | |
| Fe | | 1.70 | 1.70 | | | 0.28 | 0.31 |
| Mg | | 0.72 | 0.69 | | | 0.66 | 0.57 |
| Mn | | 0.06 | 0.05 | | | | |
| Ca | | 0.55 | 0.61 | 0.00 | 0.00 | 0.80 | 0.84 |
| Na | | | | 0.06 | 0.01 | 0.14 | 0.17 |
| K | | | | 0.95 | 0.99 | | |
| total | | 3.03 | 3.05 | 1.01 | 1.00 | 1.89 | 1.89 |

**Table B3: Representative analysis for sample F6. m=measured; c=calculated from Perple_X at 1.17 GPa and 590 °C; $Fe_2O_3$ calculated on the basis of volume per cent of phases; Solution models Gt(WPH), IlHm(A), MtUl(A), Omph(HP), GlTrTsPg, melt(HP), Chl(HP), Sp(HP), Opx(HP), Mica(CHA1), Ctd(HP), St(HP), Bio(TCC), hCrd Sapp(HP), Osm(HP), Carp(HP), Sud, feldspar, Neph(FB)**

**Appendix C**

| S5 | 26.3082 S, 131.7745 E |
|---|---|
| F6 | 26.3528 S, 131.8419 E |
| F31 | 26.2793S, 131.4968 E |
| F44 | 26.4514 S, 131.9553 E |
| F68 | 26.3849 S, 131.7067 E |
| F71 | 26.3550 S, 131.8432 E |

**Table C1 Summary of coordinates (WGS 84) of sample locations discussed in the text.**

## Author contribution

All authors listed took part in at least two of the three field seasons, which formed the basis of this study. AC's previous knowledge of the field area and the local people was essential for the success of the campaign. SW contributed to the microprobe work. NM and GP developed the initial idea of the study and the project was financed by a Swiss National Science Foundation (SNF) Grant awarded to NM. FH prepared the manuscript with contributions from all co-authors.

## Acknowledgements

We want to thank Torgeir B. Andersen and Uwe Altenberger for their thorough and critical reviews. We gratefully acknowledge permission granted to work on the Anangu Pitjantjatjara Yankunytjatjara Lands (APY) to carry out our field work in the area. The Northern Territory Geological Survey (NTGS) and Basil Tikoff (Department of Geoscience, University of Wisconsin) are thanked for their logistical support and the Nicolle family of Mulga Park station for their hospitality. The Scientific Center for Optical and Electron Microscopy (ScopeM) provided the facilities for the SEM work, and help by Karsten Kunze is especially acknowledged. The EMPA work was supported by Eric Reusser and Lukas Martin. This project was financed by the Swiss National Science Foundation (SNF) Grant 200021_146745 and by the University of Padova (BIRD175145/17: The geological record of deep earthquakes: the association pseudotachylyte-mylonite).

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
