# Peer review of "Pseudotachylyte as field evidence for lower crustal earthquakes 1 during the intracontinental Petermann Orogeny (Musgrave 2 Block, Central Australia) 3"

_Solid Earth, 2017_

## Referee Comment (RC1) · Dr. Andersen (Referee) · 21 Nov 2017

Review: Pseudotachylyte as field evidence for lower crustal earthquakes during the intracontinental Petermann Orogeny (Musgrave Block, Central Australia) By Friedrich Hawemann et al.

This paper for Solid Earth describes lower crustal, intra-continental pseudotachylytes developed at high-grade metamorphic conditions from the Musgrave Block in Australia. These rocks have been described and discussed in several previous papers (which are

adequately referred to in the manuscript). After reading this new manuscript I am still in doubt about why the pseudotachylytes (PST) are there, how they are related to the co-facial mylonitic shear zones and how they were formed. The main conclusion of the authors of this manuscript is that: "Repeated episodes of brittle failure and ductile creep represent recurring earthquake cycles and a strong variation of stress in a water deficient lower crust". You have not shown that they are cyclic, to me the changes from brittle to ductile and back again seem random (not cyclic). How these repeated 'strong variations' in stresses are formed, and the causal link between stress variations and the formation of shear zones vs. co-seismic faulting is not discussed or explained in detail in the manuscript. So, for new readers including me, their presence is still 'mysterious'. The authors reject both dehydration embrittlement, fluid infiltration weakening and 'shear-heating runaway' as weakening mechanisms to trigger co-seismic faulting. In the end, we have no real explanation for the observed phenomenon. Self-localised-thermal runaway (SLTR) following John et al (2009) is plainly rejected as a weakening mechanism in this manuscript because the authors have not found ductile precursors to any of the studied faults. I wonder if they have looked well enough? because there is no detailed description or illustration of fault veins in their figures included here. The deep crustal PST examples I have detailed knowledge from (in Corsica and Norway) we have spent a long time looking and dedicated sampled fault veins (not the nice big injection veins) to observe what happens with wall-rocks during co-seismic faults. Particularly the smallest fault veins (see Andersen et al 2008, Deseta et al. 2014) provide the best examples of ductile wall-rock damage zones. The evidence for crystal-plastic and ductile deformation is not easily found because the high heat tends to melt and destroy the evidence for the ductile wall-rock precursor as well as most of the inclusions of the wall rocks. Therefore, only a few examples provide macroscopic evidence for pre-fault (PST) ductile fabrics, one is from the Kråkeneset gabbro described in John et al. (2009) and I enclose a field photo of this for your inspection, where shear fabrics are preserved along a small fault next to a PST where they are mostly melted away on the same fault. Evidence from minor fault in thin section are more common.Therefore,

if you still find no evidence of shearing after new inspection of wall-rock damage zones in you fault veins, you are at least be able to say with confidence that evidence of SLTR is not found after careful inspection! Otherwise, perhaps you should keep an open mind to SLTR as an option until you can document that there is no crystal plasticity or ductility anywhere in the wall-rock damage zones along your fault veins.

The message of this manuscript is therefore a mainly a further documentation of the high- grade PSTs. It presents a somewhat improved determination of the metamorphic conditions during their formation by characterising the quench-mineralogy by using pseudosections and the Matlab toolbox XMapTools to calculate bulk compositions from small regions within the PSTs from wavelength dispersive spectrometer (WDS) maps. This useful and may give better constraints than bulk-rock analyses, but are also hampered by the selective masking of some minerals and mineral inclusions from the calculations. In the PST in Holsnøy described by Austrheim and co-workers, mineral inclusions in for example garnet is very commonly associated with the shock-type deformation (partial pulverisation of wall-rocks) of minerals during the co-seismic faulting, and should therefore perhaps be included? On the other hand, there may be very local grain-scale disequilibrium due to the isolation of inclusions inside minerals from the matrix.

Figures! In this part of my review with general comments I suggest that you improve most of your figures or at least the explanation in the figure text. If you discuss more in text I want you to specify where this can be found in the main text. I also want to see micrograph of fault-veins and I want better (in fact much improved) text to most of the figures. In many cases texts are very short and do not explain well enough what we see particularly in the photo figures. There are also some errors for examples in Fig 8b where the pressure unit is written as GPa but probably given in numbers as kbar? A regional geological map (Fig. 2) should normally have a regional cross-section as well. In Fig. 3 you have some nice PST images, but again the explanations in the fig-text is very short and inadequate. I miss a much better explanation of what I see

in fig 3a and 3c, and a discussion/explanation of how rotation of clasts in 3a occurred, and if there is a PST fault vein along the contact with the amphibolite dolerite and the duplex-like structure in 3b. This can be done better! And I want image(s) fault-vein contacts with wall-rocks. In figure 4 there is an inset backscatter image of an obliquely foliated injection vein? Explain what we see and why is there a foliation there. Is this flow foliation or some post PST deformation phenomenon? See comments in the manuscript text on figure 5!

Concluding remarks: My conclusions are that I would like to see this research published but that you need to improve the text and the figures and perhaps add more illustrations of what you have observed from these impressive and enigmatic PSTs. You have demonstrated beyond doubt that the co-seismic faulting took place at depth in the lower crust! The runaway weakening mechanism is, however, still not explained but you can obviously provide more information and discuss this in more detail, and perhaps use some of the papers mentioned above that you have not referred to. (The Austrheim 2013 paper is missing from the ref list.)

Comments in the Manuscript pdf file: I have made a number of comments in the manuscript-pdf and to the figures (in red). I have also highlighted some of parts of the text and figures in yellow, mostly for my own reading, but some of the red comments are directly related to the yellow highlighted text, so take a good look at these parts and see if improvements are needed.

As a conclusion, I would like to see this manuscript published but only after some careful revision; where at least some of the points I have raised are improved. I therefore recommend a thorough revision before the paper is published.

Best regards, Torgeir B. Andersen

Please also note the supplement to this comment:
https://www.solid-earth-discuss.net/se-2017-123/se-2017-123-RC1-supplement.pdf

**Supplement:**

[Figure]

Shear
fabric

PST

---

## Short Comment (SC1) · 7 Dec 2017

General comment on pseudotachylyte generation in lower crustal rocks in response to Torgeir Andersen's review of Hawemann et al. by Neil Mancktelow, Giorgio Pennacchioni and Friedrich Hawemann

Torgeir Andersen raised the fundamental point in his review about potential models for development of pseudotachylytes, and implicitly intermediate depth earthquakes, under conditions typical of the middle to lower crust. He noted that we favoured a brittle

fracture precursor to both shear zone and the pseudotachylyte development but that we did not discuss this in any detail. This is correct because the aim of the current manuscript was mainly to unequivocally establish the conditions of pseudotachylyte formation as part of a logical progression of papers where in the next submission the interaction between multiple events of fracturing, localized ductile shearing and pseudotachylyte development could be discussed with reference to the current work and without having to re-establish the conditions all over again. The crucial result of the current work is that the conditions of both shear zone and pseudotachylyte development were around 650°C and 1.2 GPa and that there is no evidence of water-rich fluid infiltration before, during or after individual periods of shear zone localization or pseudotachylyte generation. Torgeir took issue with our use of the word "cycles", but the important point is that there is repeated shearing and pseudotachylyte formation, as shown by overprinting relationships. According to John et al (2009), this is exactly what is not observed in the Krakenes gabbro, which corresponds to the photo uploaded by Torgeir. We will wait with the formal reply to Torgeir's review until we have the second review as well, so that we can update our manuscript to include both sets of comments. Most of the formal comments we can readily include and we thank Torgeir for the suggestions. However, here, in the interactive discussion section, is a good chance to discuss the general question of deep pseudotachylyte development, where we think the arguments have become a bit dogmatic and need some critical re-evaluation. Seismic frictional melting is generally accepted as a viable mechanism for pseudotachylyte generation. The perceived problem is that the differential stress required to cause brittle fracturing in deep and dry rocks, as established for the current study, is high and commonly considered to be "unrealistic". The compilation of Byerlee reaches to the range of ca. 1.2 GPa appropriate to our study and would indicate that the required differential stress (sigma1 – sigma3) should be approximately equal to the (effective) confining pressure, i.e. ca. 1.2 GPa. This "problem" of the large stresses necessary for fracture at such depths led to the proposal of two alternative mechanisms (1) dehydration embrittlement and (2) shear instabilities and "self-localizing thermal runaway".

In the case in question here, the first possibility is clearly not relevant, because the conditions are "dry" and remain "dry". So the discussion is restricted to either slip and frictional heating on a discrete fracture or crystal plastic shear localization and shear heating leading to thermal runaway. In both cases, the fundamental driving mechanism is the transfer of recoverable stored elastic energy in a larger body of surrounding rock via localized permanent deformation into heat at a rate that is faster than that at which the heat can diffuse away. The theoretical basis for self-localizing thermal runaway have been a series of numerical models from Kelemen and Hirth (2007), Braeck and Podladchikov (2007), John et al (2009), and Thielmann et al. (2015). All of these models are fundamentally 1-D, which means they assume the initial required perturbation and the shear zone that develops is planar and infinite in 2D. An important point in all these models is that they necessarily require an initial precursor perturbation in the rheology. In this sense they are not strictly "self-localizing" – the planar zone of localization is actually prescribed. It is accentuated during subsequent deformation but it is present from the beginning. In the case of John et al. (2009) this is justified by the statement that both the eclogite-facies shear zones and the pseudotachylytes "have higher degrees of hydration, caused by infiltration of external fluids, and up to three-orders-of-magnitude-smaller grain sizes than the almost dry wall rock". So this requires some planar precursor that has reduced grain size, increased permeability and allowed fluid infiltration. Our experience from other areas (Mancktelow and Pennacchioni 2005; Pennacchioni and Mancktelow 2007; Menegon and Pennacchioni 2009; Gonclaves et al. 2016) and from the Musgrave Block, is that this necessary initial precursor in originally relatively homogeneous and isotropic rock is itself a fracture – the question then is whether this fracture also developed under deep conditions, in which case we are back to the original argument about whether fractures can develop in deep dry rocks? The geometry of the shear zones also suggests some form of precursor fracture. If they developed under viscoelastic conditions from point irregularities a more conjugate pattern would be expected, with initial angles at 45° to the shortening direction (e.g. Grujic and Mancktelow 1998; Mancktelow 2002). The photo that Torgeir provided as

part of his review is somewhat misleading as it could be taken to imply, without further explanation, that the pseudotachylyte developed by thermal runaway from the shear zone. This clearly cannot be the case because the displacement across the ductile shear zone is too small. Alternatives would be (1) that the shear zone itself is localizing on the already existing pseudotachylyte or (2) that the pseudotachylyte localizes on the pre-existing shear zone. However, in the original John et al (2009) paper it is claimed that "both types (i.e. shear zones and pseudotachylytes) formed in a single, continuous and fast event". In this interpretation, the shear zones are cases that did not make the step to thermal runaway and the pseudotachylytes the cases that did – and supposedly consumed almost all the evidence for the shear zone of the initial stages. In Fig. 1(h and j) of that paper, this is implied by the final broadening of the zone of melting (which would indeed involve a "delocalization" and broadening in the final stage). This is then taken as an argument why the precursor ultramylonite of the shear zone is not preserved as clasts – the evidence is lost due to complete melting of the ultramylonite precursor. However, this is rather hard to believe. As can be seen in the natural example of Fig. 1b of John et al. (2009) (although too small to really see details), there are plenty of clasts within the pseudotachylyte all showing evidence for brittle fracture and not ultramylonitic shear. In the model of Thielmann et al (2015), there is instead real continued localization, so that the pseudotachylyte should be observed within a broader shear zone. Our experience from the Musgrave Block is that the clasts directly reflect the protolith in which the pseudotachylyte developed. If this protolith was little deformed, then the clasts appear to be generally brittle without mylonitization. If, as is sometimes the case (see uploaded image), the pseudotachylyte developed in a pre-existing shear zone, then the clasts are also directly comparable to the surrounding matrix and are not always totally consumed. Following the model of John et al. (2009), in this case it should be expected that the whole precursor shear zone should melt and a geometry as in the photo would not be expected. It should be noted in this photo that there is common small garnet in the background protomylonite, in the localized mylonite/ultramylonite, in the clast within the pseudotachylyte, and in

the pseudotachylyte injection veins, so that all these structures developed under the regional lower crustal conditions (i.e. ca. 650°C, 1.2 GPa). So, in summary, we do not exclude that thermal runaway in viscoelastic shear zones may occur and could explain some natural examples. However, the observational evidence from the area of the current study strongly suggests that brittle fracture is a necessary precursor for both shear zone localization and pseudotachylyte formation – with the necessary implication that differential stresses were (at least transiently) sufficiently high for brittle fracture under dry high pressure conditions.

[Figure]

**Fig. 1.**

---

## Referee Comment (RC2) · U. Altenberger (Referee) · 9 Dec 2017

Comment on "Pseudotachylyte as field evidence for lower crustal earthquakes during the intracontinental Petermann Orogeny (Musgrave Block, Central Australia)" by Friedrich Hawemann et al.

by Prof. Dr. Uwe Altenberger, University of Potsdam, Germany

The article is concentrated on the phenomenon of a multiple brittle-ductile deformation sequence in the Muscrave Block, Australia. The aim of the manuscript is to describe

the fabric and petrology of the brittle event, documented by pseudotachylytes (pst) in detail and to interpret the strong connection of brittle and seismic events. I try to read the manuscript very carefully, because I had worked on deep crustal pst, too and hoped to learn something new. On one hand, there are a lot of interesting observations on different generations oft pst in the Davenport shear zone, on the other hand some descriptions are missing or are too short to evaluate possible results and conclusions. The presented work has to my opinion two strong topics: 1. the proof of deep crustal fault-related frictional melts by petrological methods and 2. the proof of cyclic repetition of brittle and ductile processes. Both topics are managed, but not sufficiently. The manuscript has a loot of weak points, which has to be corrected before publishing. The petrographic observations of the host rocks are little and sometimes not clear but these have strong relation to the conclusion These rocks, especially the contact zone to the pst should be described in more detail. Are there any remnants of previous, possibly ultra-mylonitic, deformations? Are the pst concentrated in special layers of the protolithe, e-.g. involving more (OH) - bearing phases? In the description of the dolerite, as a protolithe, there is no given mineral assemblage (does it include grt or hbl as a (OH)-bearing phase?). Is there any thin-section or SEM image of the mylonites adjacent to the pst (e.g. a prolongation of Fig 4). Is the brittle deformation a direct consequence of the ductile deformation ?, e.g. same layers, or discordant after changing the stress system? The reader is not informed if the minerals described are "magmatic", i.e. crystallized directly from the melt or if these are formed (overprinted) by the crustal metamorphism. In the deep crustal environment this is not easy to distinguish but has a strong impact on the interpretation. We know from some places, that kyanite can crystallize from the melt and, as it is described, the garnet with cauliflower structures are a clear evidence for rapid cooling, i.e. crystallizing directly from the melt. In addition, pyroxenes can form under high-temperature or high-pressure conditions from a melt, or recrystallized later from the very fine-grained to glassy pst matrix. And how can we know, that kyanite is formed in the sample, not sillmanite? They are probably too small to distinguish by the used methods, XRD is need to confirm this, not pseudosections. A point of interest is also: which minerals from the protolithe are consumed and which are stable. I think biotite will directly melt, producing some (OH). Amphibole too. The consumed minerals control the composition and rheological behavior of the melt. The descriptions of some important figures like Fig 4 is too short-and do not describe the four generations of pst sufficiently. Some simple ideas have no base, if it is written, two generations of pst overprinted by ductile deformation are an indicator of cyclic brittle and ductile deformation. It is only an evidence for two phases of brittle deformation followed by ductile deformation. What is the PT-conditions of the ductile deformation-any evidence? Is it possible, that the ductile event is part of the retrograde exhumation? Some parts are clearly described but not well thought: a pst in a gabbro is containing Kfs clasts - gabbros should not contain Kfs. And the matrix is free of plagioclase and composed of Grt+Cpx+Kfs+Qtz? What does this mean? Is the melt travelled a longer distance, from a different protolithe? I agree with the used method of pseudosections. However, is there any further indication for the deep crustal evolution, like high $Al_2O_3$ concentration in the newly formed pyroxenes, what is the composition of the melt-derived garnets-there are experimental data on the P-conditions of garnets formed from magmatic melts (given in the cited Altenberger 2011, 2013) In addition, the word recrystallization, which is often used by people from petrology, structural geology as well as from geochemistry; is often used in a different way. Therefore, please write if in the described examples recrystallization is crystallization from the melt (e.g. grt) or recrystallized under metamorphic conditions during later times from the fine-grained matrix? This is important, although not easy to distinguish. We often have the situation that quenched crystals have a metamorphic rim etc. You can calculate by your data also the geothermal gradient - it is only ca 20°/km. So the seismicity has happened in a relatively cold crust.

Although there is no real evidence for cyclicity, there is a well-described evidence of a polyphase evolution. However, I am wondering, that the classical paper of Handy & Bruhn (2004, EPSL,223), thinking about the cyclicity and "Stress– strain evolution for a volume of rock undergoing deformation to frictional sliding or creep at a constant slip or strain rate" is not cited. The interpretations in the manuscript are not satisfying, but maybe there is no simple answer. Is there any correlation with the drastic change in shear direction from sinsitral to dextral?

I attached some additional corrections to the original manuscript.

To resume: the manuscript is worth to get published. It will submit more data to these deep crustal and not well understood lower crustal processes, although a satisfying interpretation is not given by the authors, yet.

Please also note the supplement to this comment:
https://www.solid-earth-discuss.net/se-2017-123/se-2017-123-RC2-supplement.pdf

**Supplement:**

[revised manuscript text omitted]

---

## Author Comment (AC1) · 22 Feb 2018

RC: You have not shown that they are cyclic, to me the changes from brittle to ductile and back again seem random (not cyclic). AR: With the word "cyclic" we wanted to express that deformation is changing from brittle to ductile and back. We try to demonstrate this by pointing out, that pseudotachylytes are emplaced pre-, syn- and post shearing. AC: We integrated a new Fig 6 to show that sheared pseudotachylytes can be found as clasts in a new generation of pseudotachylyte, demonstrating the switch from brittle to ductile to brittle again, which may actually repeat several times. We now

restrict the use of the term "cycles" to the discussion and specifically with reference to earthquake cycles.

RC: How these repeated 'strong variations' in stresses are formed, and the causal link between stress variations and the formation of shear zones vs. co-seismic faulting is not discussed or explained in detail in the manuscript. AR: In the discussion, we test our observations against the common proposed models for brittle-ductile interplay in the lower crust. With the data presented here, we cannot establish the cause of the stress variations. This problem will be specifically addressed in a different manuscript, which is currently in preparation. The aim of the current manuscript is different and quite specific – to establish that repeated cycles from brittle to ductile to brittle, involving large volumes of pseudotachylyte, are occurring under water deficient conditions of ca. 650 °C and 1.2 GPa, i.e. lower crustal conditions

RC: Self-localised thermal runaway (SLTR) following John et al (2009) is plainly rejected as a weakening mechanism in this manuscript because the authors have not found ductile precursors to any of the studied faults. I wonder if they have looked well enough? because there is no detailed description or illustration of fault veins in their figures included here. The deep crustal PST examples I have detailed knowledge from (in Corsica and Norway) we have spent a long time looking and dedicated sampled fault veins (not the nice big injection veins) to observe what happens with wall-rocks during co-seismic faults. Particularly the smallest fault veins (see Andersen et al 2008, Deseta et al. 2014) provide the best examples of ductile wall-rock damage zones. The evidence for crystal-plastic and ductile deformation is not easily found because the high heat tends to melt and destroy the evidence for the ductile wall-rock precursor as well as most of the inclusions of the wall rocks. Therefore, only a few examples provide macroscopic evidence for pre-fault (PST) ductile fabrics, one is from the Kråkeneset gabbro described in John et al. (2009) and I enclose a field photo of this for your inspection, where shear fabrics are preserved along a small fault next to a PST where they are mostly melted away on the same fault. Evidence from minor fault in thin section are more common.Therefore, if you still find no evidence of shearing after new inspection of wall-rock damage zones in you fault veins, you are at least be able to say with confidence that evidence of SLTR is not found after careful inspection! Otherwise, perhaps you should keep an open mind to SLTR as an option until you can document that there is no crystal plasticity or ductility anywhere in the wall-rock damage zones along your fault veins. [. . .] And I want image(s) fault-vein contacts with wall-rocks. AR: We tried to have an unbiased view with regard to the formation mechanism of the pseudotachylytes in the Musgrave Block. The observations we made are in conflict with the idea of SLTR. In the new Fig. 6, we present a pseudotachylyte fault vein including the host rock, where no ultramylonitic precursor is visible. A more thorough discussion is presented in the short comment (SC1) in the discussion. We did not add a more thorough discussion to the current manuscript, as it is beyond the scope and aim of this study.

RC: In the PST in Holsnøy described by Austrheim and co-workers, mineral inclusions in for example garnet is very commonly associated with the shock-type deformation (partial pulverisation of wall-rocks) of minerals during the co-seismic faulting, and should therefore perhaps be included? AR: We do report fractured garnets, but they show discrete and rather planar fractures and are not pulverized. AC: The connection between fractured garnet and seismic stresses is now added to the text, together with relevant references.

RC: Figures! In this part of my review with general comments I suggest that you improve most of your figures or at least the explanation in the figure text. If you discuss more in text I want you to specify where this can be found in the main text. I also want to see micrograph of fault-veins and I want better (in fact much improved) text to most of the figures. In many cases texts are very short and do not explain well enough what we see particularly in the photo figures. There are also some errors for examples in Fig 8b where the pressure unit is written as GPa but probably given in numbers as kbar? [. . .] In Fig. 3 you have some nice PST images, but again the explanations in the fig-text is

very short and inadequate. I miss a much better explanation of what I see in fig 3a and 3c, and a discussion/explanation of how rotation of clasts in 3a occurred, and if there is a PST fault vein along the contact with the amphibolite dolerite and the duplex-like structure in 3b. This can be done better! AC: Figure captions have been improved to aid a better understanding of the images. However, in principle, we consider that figure captions should be concise and limited to description rather than interpretation. The figures are all described and discussed in detail within the main body of the text. We apologize for the error made in Fig 8b, which has been corrected.

RC: A regional geological map (Fig. 2) should normally have a regional cross-section as well. AR: In our opinion, the geophysical maps are more instructive for the purpose than a geological map, as these also see through the cover providing a clearer tectonic interpretation and highlighting the difference between Mulga Park- and Fregon Subdomains, as well as the post-Musgrave Orogeny granites that were not depleted in Th. AC: We included a recent reference (Wex et al. 2017) where a geological map and cross sections can be found.

RC: In figure 4 there is an inset backscatter image of an obliquely foliated injection vein? Explain what we see and why is there a foliation there. Is this flow foliation or some post PST deformation phenomenon? AR: As the foliation is slightly oblique to the margin of the vein, we interpret the foliation to be the result of ductile shearing. AC: This has been clarified in the text.

RC1_supplement: Other comments, if not already addressed above, have been integrated in text and figures.

RC: Silicate melts may have a high fluid, any info on the content of fluids in the pst? AR: As clearly shown in the sample description, biotite is the only water-bearing mineral observed in the studied sample (F68). Furthermore, kyanite rather than clinozoisite/epidote is present in the pseudotachylyte, also indicating that the fluid content of the initial melt was low.

RC: injection vein with later localisation? (comment of Fig. 3b) AR: The geometries are not clear enough, to call this an injection vein. The ductile overprint is later, as stated in the text.

RC: is there any evidence for quenched mineral zoning as evidence for progressive cooling? AR: The grain size of minerals is extremely small, and no zoning is visible.

RC: Do we see two generations of pst or just a transition from foliated to not foliated? Text is not adequate for reader to understand this brittle- to ductile transition. Explain better! I think more illustrations are required, also optical micrograph, to me this could look as a ductile precursor to a static quenched pst! (comment to Fig.5a) AR: The host rock is a quartzo-feldspathic mylonite, with clearly visible quartz ribbons (appear dark). We here want to demonstrate the brittle overprint of the mylonitic foliation. There is no evidence, that the mylonite represents a ductile precursor. Also, in the model of John et al. (2009), the precursor (ultra-) mylonite is expected to be completely melted. AC: The figure captions have been improved to clarify this.

RC: Is there any issue between cooling from max shear heating (friction) temperatures and temperatures derived from mineral equilibria modelling? Could there have been superheating? AR: As the temperatures are derived from equilibria modelling of dynamically recrystallized minerals and not on minerals directly crystallizing from a melt, we are confident that superheating effects are not reflected in the mineral compositions.

RC: P-1.2 GPa and 690C are considerably more narrow than the pseudosection shows, any reason for this? AR: The range of this field is very narrow anyways, so for simplicity the center point is used as input to calculate the mineral composition. We do not claim the method to be this accurate.

RC: Mineral inclusion masked from PT models because they are considered not to be part of a stable assemblage, is this justified, we see crystals f.example garnet as 'sponges' of inclusion due to seismic deformation. see Austrheim papers AR: We did not mask mineral inclusions. We masked clasts that were not dynamically recrystallized

and therefore not part of the stable assemblage.

RC: 'close to these conditions', is there a ref. for this? AR: This is derived from the pseudosection.

RC: insufficient explanation of element maps, mineral names on fig required, what is blue in Fe in C. AR: The initial idea of the element maps was to show the reaction of feldspar clasts and the different iron-oxide phases. We agree, that a larger map with labeled phases can be helpful for the reader. AC: An enlarged version of Fig. 9a with mineral labels will be added to the appendix.

---

## Author Comment (AC2) · 22 Feb 2018

RC (referee comment): The petrographic observations of the host rocks are little and sometimes not clear but these have strong relation to the conclusion AC (author's changes to the manuscript): The petrographic descriptions of the host rock have been added and additional references provided, including a recent one by Wex et al (2017) from the same research group where the host rock conditions are considered in more detail..

RC: Are there any remnants of previous, possibly ultra-mylonitic, deformations? AR: This was already addressed in the manuscript by clearly separating the pseudotachylytes into three categories with respect to ductile shearing– so yes, there are examples where the pseudotachylyte post-dates strong shearing. However, as visible in Fig. 3c, there are also examples of pseudotachylytes that can be found in undeformed host rocks.

RC: Are the pst concentrated in special layers of the protolithe, e-.g. involving more (OH) - bearing phases? AR: The OH-bearing minerals are mostly limited to late- to post Musgravian intrusions. There is no affinity of pseudotachylytes to these lithologies. AC: This has been added to the field observations.

RC: In the description of the dolerite, as a protolith, there is no given mineral assemblage (does it include grt or hbl as a (OH)-bearing phase?). AR: The assemblage of the dolerites is "dry". AC: The description of the paragenesis has been added to the description of sample S5.

RC: Is there any thin-section or SEM image of the mylonites adjacent to the pst (e.g. a prolongation of Fig 4). AR: The sample of Fig. 4 is a pseudotachylyte breccia in an undeformed host rock. Examples for the ductile shearing can be found in Fig. A2 and Fig. 5a.

RC: Is the brittle deformation a direct consequence of the ductile deformation ?, e.g. same layers, or discordant after changing the stress system? AR: Pseudotachylytes emplaced in mylonites often show localization on foliation planes, as seen for example in Fig. 3b, new Fig. 5 and Fig. A2 and in the Fig 1 of the short comment. However, the opposite can also be found as late stage pseudotachylytes crosscut the mylonitic foliation and have a somewhat random orientation. AC: This was clarified in section 3.

RC: The reader is not informed if the minerals described are "magmatic", i.e. crystallized directly from the melt or if these are formed (overprinted) by the crustal metamorphism. In the deep crustal environment this is not easy to distinguish but has a strong

impact on the interpretation. We know from some places, that kyanite can crystallize from the melt and, as it is described, the garnet with cauliflower structures are a clear evidence for rapid cooling, i.e. crystallizing directly from the melt. AR: In the samples we used to derive the metamorphic conditions, the minerals are thought to form during dynamic recrystallization of the pseudotachylyte. In sample F44, for example, generation 1 remains unsheared and the minerals might well have crystallized directly from the melt or represent a static overgrowth of the former melt. However, the cauliflower garnet in Fig. 4 overgrows a planar foliation resulting from ductile shearing. We therefore argue that in this case the cauliflower garnet is not crystallized directly from the melt. The cauliflower garnets in the Fig. 5c however, can well be the result of direct crystallization from the melt, as they are hosted in an unsheared pseudotachylyte. AC: Fig. 4 was extended to clearly emphasize the difference between the pseudotachylyte generations. The text was modified to clearly state whether the minerals grew from the melt, statically or by dynamic recrystallization.

RC: And how can we know, that kyanite is formed in the sample, not sillmanite? They are probably too small to distinguish by the used methods, XRD is need to confirm this, not pseudosections. AR: Pseudosections had not been used to identify minerals in the thin sections. Kyanite was distinguished from sillimanite by using Raman spectroscopy and EBSD. AC: This information has been added to the text.

RC: A point of interest is also: which minerals from the protolithe are consumed and which are stable. I think biotite will directly melt, producing some (OH). AR: In the example of sample F68, biotite is slightly enriched in the pseudotachylyte. However, the amount of OH produced is small, as no new OH-bearing phases are found in the pseudotachylyte assemblage. Garnet is also readily molten, as it never appears as clasts in the pseudotachylytes. Quartz is commonly found as clasts, for example visible in Fig. 5a, where whole ribbons of quartz "survive" the melt formation. In the example of F6, most clasts are made up of plagioclase. AC: This information was integrated into the manuscript.

RC: The descriptions of some important figures like Fig 4 is too short-and do not describe the four generations of pst sufficiently. AC: The description of figures where enlarged, Fig. 4 was augmented with further backscatter images for all generations of pseudotachylytes.

RC: Some simple ideas have no base, if it is written, two generations of pst overprinted by ductile deformation are an indicator of cyclic brittle and ductile deformation. It is only an evidence for two phases of brittle deformation followed by ductile deformation. AC: A new Figure 6 was added, to demonstrate the switch from brittle to ductile deformation and back to brittle. It is true, that this only represents one cycle, from brittle to ductile and back to brittle, but the chances of preservation of multiple cycles are low. In the new version of the manuscript, we avoid the use of the word "cyclic" and restrict the use of the word to the discussion part.

RC: What is the PT-conditions of the ductile deformation-any evidence? Is it possible, that the ductile event is part of the retrograde exhumation? AR: The ductile deformation in the Davenport Shear Zone is described in detail Camacho et al. (1997), as stated in the text. As the mylonites host the sub-eclogitic assemblage, we can exclude a ductile retrograde overprint.

RC: Some parts are clearly described but not well thought: a pst in a gabbro is containing Kfs clasts - gabbros should not contain Kfs. AR: There are no clasts of Kfs, but Pl-clasts are overgrown by Kfs. AC: This error has been corrected in the text.

RC: I agree with the used method of pseudosections. However, is there any further indication for the deep crustal evolution, like high $Al_2O_3$ concentration in the newly formed pyroxenes, what is the composition of the melt-derived garnets-there are experimental data on the P-conditions of garnets formed from magmatic melts. AR: The pseudotachylyte-melt derived minerals are for sure interesting in many ways, but the study of those would be beyond the scope of this publication. As garnets crystallize from the pseudotachylyte melt, they probably record the temperature of the melt, which

is much higher than the ambient conditions. Al-rich pyroxenes have been described from pseudotachylytes in the Musgrave Ranges by Wenk and Weiss (1982), and the applied barometers return pressures of about 3 GPa, thought to represent dynamic pressures, rather than lithostatic. AC: The comment about the pyroxenes have been added to section 4.1.

RC: Therefore, please write if in the described examples recrystallization is crystallization from the melt (e.g. grt) or recrystallized under metamorphic conditions during later times from the fine-grained matrix? AC: This has been clarified in the text.

RC: You can calculate by your data also the geothermal gradient - it is only ca 20 °/km. AC: This information has been added to the results.

RC: However, I am wondering, that the classical paper of Handy & Bruhn (2004, EPSL,223), thinking about the cyclicity and "Stress– strain evolution for a volume of rock undergoing deformation to frictional sliding or creep at a constant slip or strain rate" is not cited. AC: The work of Handy and Brun is now cited.

RC: The interpretations in the manuscript are not satisfying, but maybe there is no simple answer. AR: The model of downward propagation of seismic stresses from the upper crust is favoured by many authors in recent publications, and is physically feasible but in our opinion there is no unequivocal geological evidence in previous publications that exclusively support this model. We therefore wanted to critically evaluate this model in the current study and to show the contradictions. This highlights rather than "solves" the problem and if anything provides an impetus for future studies.

RC: Is there any correlation with the drastic change in shear direction from sinsitral to dextral? AR: The change from sinistral to dextral sense of shear is more likely the result of slight variation in orientation of the shear zones, as described in the text, as this change is lateral in space and not temporal. In some cases, shear zones do show a change in sense of shear, but no consistent change can be documented.

RC2_supplement: RC: Nice maps, but a map of the local geology, where the samples are taken from would be of interest, too. AR: The geological background is kept concise, therefore we did not include a geological map or profile. The geophysical maps provide a direct insight through the cover, and help to identify the main shear zones as well as the difference in metamorphic grade. AC: We modified the text to indicate better where a geological map and cross section can be found.

RC: I would add the beautiful BSE image A 1 from the Appendix with the flow folds, which is not described in the text, yet. AC:. We integrated the beautiful image A1 into figure 5 along with a better description.

RC2_supplement: Other comments, if not already addressed above, have been integrated in text and figures.

RC: why is it sheared? Give an evidence- AR: There is an internal foliation visible, which is defined by garnet and biotite. This is stated in the text.

RC: And the red box is to boarder of the red box are too thin-better to do this in white AC: The outline of the red box is now thicker.

All other comments from the supplement are discussed above. Figure captions have been enhanced to provide better insight.